# Variational Policy Gradient Method for Reinforcement Learning with General Utilities

**Junyu Zhang**
Department of Electrical Engineering
Center for Statistics and Machine Learning
Princeton University, Princeton, NJ 08544
`junyuz@princeton.edu`

**Alec Koppel**
CISD
US Army Research Laboratory
Adelphi, MD 20783
`alec.e.koppel.civ@mail.mil`

**Amrit Singh Bedi**
CISD
US Army Research Laboratory
Adelphi, MD 20783
`amrit0714@gmail.com`

**Csaba Szepesvári**
Department of Computer Science
DeepMind/University of Alberta
`szepesva@ualberta.ca`

**Mengdi Wang**
Department of Electrical Engineering
Center for Statistics and Machine Learning
Princeton University/Deepmind, Princeton, NJ 08544
`mengdiw@princeton.edu`

## Abstract

In recent years, reinforcement learning (RL) systems with general goals beyond a cumulative sum of rewards have gained traction, such as in constrained problems, exploration, and acting upon prior experiences. In this paper, we consider policy optimization in Markov Decision Problems, where the objective is a general concave utility function of the state-action occupancy measure, which subsumes several of the aforementioned examples as special cases. Such generality invalidates the Bellman equation. As this means that dynamic programming no longer works, we focus on direct policy search. Analogously to the Policy Gradient Theorem [44] available for RL with cumulative rewards, we derive a new Variational Policy Gradient Theorem for RL with general utilities, which establishes that the parametrized policy gradient may be obtained as the solution of a stochastic saddle point problem involving the Fenchel dual of the utility function. We develop a variational Monte Carlo gradient estimation algorithm to compute the policy gradient based on sample paths. We prove that the variational policy gradient scheme converges globally to the optimal policy for the general objective, though the optimization problem is nonconvex. We also establish its rate of convergence of the order $O(1/t)$ by exploiting the hidden convexity of the problem, and proves that it converges exponentially when the problem admits hidden strong convexity. Our analysis applies to the standard RL problem with cumulative rewards as a special case, in which case our result improves the available convergence rate.

## 1   Introduction

The standard formulation of reinforcement learning (RL) is concerned with finding a policy that maximizes the expected sum of rewards along the sample paths generated by the policy. The additive

nature of the objective function creates an attractive algebraic structure which most efficient RL algorithms exploit. However, the cumulative reward objective is not the only one that has attracted attention. In fact, many alternative objectives made appearances already in the early literature on stochastic optimal control and operations research. Examples include various kinds of risk-sensitive objectives [23, 10, 52, 28], objectives to maximize the entropy of the state visitation distribution [18], the incorporation of constraints [14, 3, 1], and learning to "mimic" a demonstration [39, 4]. Maximum entropy search as it currently exists alternates between density estimation and a planning oracle [18], whose operation is restricted to tabular settings. Special cases of that which proposed here alleviate the need for a planning oracle and moreover incorporate policy parameterizations which are a prerequisite for large spaces. Also worth mentioning is Online Relative Entropy Policy Search (O-REPS)[56], which considers entropy-defined trust regions to ensure stable policy adaptation between off and on policy search. Doing so, however, is categorically different from maximizing the entropy of a policy itself: rather than safely adapting a policy, one seeks a policy capable of covering an unknown space quickly.

In this paper, we consider RL with general utility functions, and we aim to develop a principled methodology and theory for policy optimization in such problems. We focus on utility functions that are concave functionals of the state-action occupancy measure, which contains many, although not all, of the aforementioned examples as special cases.The general (or non-standard [23]) utility is a strict generalization of cumulative reward, which itself can be viewed as a linear functional of the state-action occupancy measure, and as such, is a concave function of the occupancy measures.

When moving beyond cumulative rewards, we quickly run into technical challenges because of the lack of additive structure. Without additivity of rewards, the problem becomes non-Markovian in the cost-to-go [45, 49]. Consequently, the Bellman equation fails to hold and dynamic programming (DP) breaks down. Therefore, stochastic methods based upon DP such as temporal difference [43] and Q-learning [48, 37] are inapplicable. The value function, the core quantity for RL, is not even well defined for general utilities, thus invalidating the foundation of value-function based approach to RL.

Due to these challenges, we consider direct policy search methods for the solution of RL problems defined by general utility functions. We consider the most elementary policy-based method, namely the Policy Gradient (PG) method [50]. The idea of policy gradient methods is that to represent policies through some policy parametrization and then move the parameters of a policy in the direction of the gradient of the objective function. When (as typical) only a noisy estimate of the gradient is available, we arrive at a stochastic approximation method [36, 24]. In the classical cumulative reward objectives, the gradient can be written as the product of the action-value function and the gradient of the logarithm of the policy, or policy score function [44]. State-of-the-art RL algorithms for the cumulative reward setting combine this result with other ideas, such as limiting the changes to the policies [38, 40, 41], variance reduction [20, 32, 51], or exploiting structural aspects of the policy parametrization [47, 2, 29].

As mentioned, these approaches crucially rely on the standard PG Theorem [44], which is not available for general utilities. Compounding this challenge is the fact that the action-value function is not well-defined in this instance, either. Thus, how and whether the policy gradient can be effectively computed becomes a question. Further, due to the problem's nonconvexity, it is an open question whether an iterative policy improvement scheme converges to anything meaningful: In particular, while standard results for stochastic approximation would give convergence to stationary points [9], it is unclear whether the stationary points give reasonable policies. Therefore, we ask the question:

*Is policy search viable for general utilities,*
*when Bellman's equation, the value function, and dynamic programming all fail?*

We will answer the question positively in this paper. Our contributions are three-folded:

- We derive a Variational Policy Gradient Theorem for RL with general utilities which establishes that the parameterized policy gradient is the solution to a stochastic saddle point problem.

- We show that the Variational Policy Gradient can be estimated by a primal-dual stochastic approximation method based on sample paths generated by following the current policy [5]. We prove that the random error of the estimate decays at order $O(1/\sqrt{n})$ that also depends on properties of the utility, where $n$ is the number of episodes .

- We consider the non-parameterized policy optimization problem which is nonconvex in the policy space. Despite the lack of convexity, we identify the problem's hidden convexity, which allows us to show that a variational policy gradient ascent scheme converges to the global optimal policy for general utilities, at a rate of $O(1/t)$, where $t$ is the iteration index. In the special case of cumulative rewards, our result improves upon the best known convergence rate $O(1/\sqrt{t})$ for tabular policy gradient [2], and matches the convergence rate of variants of the algorithm such as softmax policy gradient [29] and natural policy gradient [2]. In the case where the utility is strongly concave in occupancy measures (e.g., utilities involving Kullback-Leiber divergence), we established the exponential convergence rate of the variational gradient scheme.

**Related Work.** Policy gradient methods have been extensive studied for RL with cumulative returns. There is a large body of work on variants of policy-based methods as well as theoretical convergence analysis for these methods. Due to space constraints, we defer a thorough review to Supplement A.

**Notation.** We let $\mathbb{R}$ denote the set of reals. We also let $\|\cdot\|$ denote the 2-norm, while for matrices we let it denote the spectral norm. For the $p$-norms ($1 \leq p \leq \infty$), we use $\|\cdot\|_p$. For any matrix $B$, $\|B\|_{\infty,2} := \max_{\|u\|_\infty \leq 1} \|Bu\|_2$. For a differentiable function $f$, we denote by $\nabla f$ its gradient. If $f$ is nondifferentiable, we denote by $\hat{\partial} f$ the Fréchet superdifferential of $f$; see e.g. [15].

## 2 Problem Formulation

Consider a Markov decision process (MDP) over the finite state space $\mathcal{S}$ and a finite action space $\mathcal{A}$. For each state $i \in \mathcal{S}$, a transition to state $j \in \mathcal{S}$ occurs when selecting action $a \in \mathcal{A}$ according to a conditional probability distribution $j \sim \mathcal{P}(\cdot|a,i)$, for which we define the short-hand notation $P_a(i,j)$. Let $\xi$ be the initial state distribution of the MDP. We let $S$ denote the number of states and $A$ the number of actions. The goal is to prescribe actions based on previous states in order to maximize some long term objective. We call $\pi : \mathcal{S} \to P(\mathcal{A})$ a *policy* that maps states to distributions over actions, which we subsequently stipulate is stationary[1]. In the standard (cumulative return) MDP, the objective is to maximize the expected cumulative sum of future rewards [35], i.e.,

$$\max_\pi V^\pi(s) := \mathbb{E}\left[\sum_{t=0}^\infty \gamma^t r_{s_t a_t} \,\middle|\, i_0 = s, a_t \sim \pi(\cdot|s_t), t = 0, 1, \dots\right], \quad \forall s \in \mathcal{S}. \tag{1}$$

with reward $r_{s_t a_t} \in \mathbb{R}$ revealed by the environment when action $a_t$ is chosen at state $s_t$.

In this paper we consider policy optimization for maximizing general objective functions that are not limited to cumulative rewards. In particular, we consider the problem

$$\max_\pi R(\pi) := F(\lambda^\pi) \tag{2}$$

where $\lambda^\pi$ is known as the *cumulative discounted state-action occupancy measure*, or *flux* under policy $\pi$, and $F$ is a general concave functional. Denote $\Delta_\mathcal{A}^\mathcal{S}$ and $\mathcal{L}$ as the set of policy and flux respectively, then $\lambda^\pi$ is given by the mapping $\Lambda : \Delta_\mathcal{A}^\mathcal{S} \mapsto \mathcal{L}$ as

$$\lambda_{sa}^\pi = \Lambda_{sa}(\pi) := \sum_{t=0}^\infty \gamma^t \cdot \mathbb{P}\Big(s_t = s, a_t = a \,\Big|\, \pi, s_0 \sim \xi\Big) \quad \text{for} \quad \forall a \in \mathcal{A}, \forall s \in \mathcal{S}. \tag{3}$$

Similar to the LP formulation of a standard MDP, we can write (2) equivalently as an optimization problem in $\lambda$ (see [53]), giving rise to

$$\max_\lambda F(\lambda) \quad \text{s.t.} \quad \sum_{a \in \mathcal{A}} (I - \gamma P_a^\top)\lambda_a = \xi, \lambda \geq 0, \tag{4}$$

where $\lambda_a = [\lambda_{1a}, \cdots, \lambda_{Sa}]^\top \in \mathbb{R}^A$ is the $a$-th column of $\lambda$ and $\xi$ is the initial distribution over the state space $\mathcal{S}$. The constraints require that $\lambda$ be the unnormalized state-action occupancy measure corresponding to *some* policy. In fact, it is well known that a policy $\pi$ inducing $\lambda$ can be extracted

from $\lambda$ using the mapping $\Pi : \mathcal{L} \mapsto \Delta_{\mathcal{A}}^{\mathcal{S}}$ as $\pi(a|s) = \Pi_{sa}(\lambda) := \frac{\lambda_{sa}}{\sum_{a' \in \mathcal{A}} \lambda_{sa'}}$ for all $a, s$. Problem (2) contains the original MDP problem as a special case. To be specific, when $F(\lambda) = \langle r, \lambda \rangle$ with $r \in \mathbb{R}^{SA}$ as the reward function, then $F(\lambda) = \langle \lambda, r \rangle = \mathbb{E}\left[\sum_{t=0}^{\infty} \gamma^t r_{s_t a_t} \mid \pi, s_0 \sim \xi\right]$. This means that (4) is a generalization of (1), and reduces to the dual LP formulation of standard MDP for this (linear) choice of $F(\cdot)$ [22]. We focus on the case where $F$ is concave, which makes (4) a concave (hence, convenient) maximization problem. Next we introduce a few examples that arise in practice for incentivizing safety, exploration, and imitation, respectively.

**Example 2.1** (**MDP with Constraints or Barriers**). In discounted constrained MDPs the goal is to maximize the total expected discounted reward under a constraint where for some cost function $c : \mathcal{S} \times \mathcal{A} \to \mathbb{R}$, the total expected discounted cost incurred by the chosen policy is constrained from above. Let $r$ denote the reward function over $\mathcal{S} \times \mathcal{A}$, the underlying optimization problem becomes

$$\max_{\pi} v_r^{\pi} := \mathbb{E}^{\pi}\left[\sum_{t=0}^{\infty} \gamma^t r(s_t, a_t)\right] \qquad \text{s.t.} \qquad v_c^{\pi} := \mathbb{E}^{\pi}\left[\sum_{t=0}^{\infty} \gamma^t c(s_t, a_t)\right] \leq C. \qquad (5)$$

As is well known, a relaxed formulation is

$$\max_{\lambda} F(\lambda) := \langle \lambda, r \rangle - \beta \cdot p(\langle \lambda, c \rangle - C) \qquad \text{s.t.} \qquad \sum_{a \in \mathcal{A}}(I - \gamma P_a^{\top})\lambda_a = \xi, \lambda \geq 0. \qquad (6)$$

where $p$ is a penalty function (e.g., the log barrier function).

**Example 2.2** (**Pure Exploration**). In the absence of a reward function, an agent may consider the problem of finding a policy whose stationary distribution has the largest "entropy", as this should facilitate maximizing the speed at which the agent explores its environment [18]:

$$\max_{\pi} R(\pi) := \text{Entropy}(\bar{\lambda}^{\pi}), \qquad (7)$$

where $\bar{\lambda}^{\pi}$ is the normalized state visitation measure given by $\bar{\lambda}_s^{\pi} = (1 - \gamma)\sum_a \lambda_{sa}^{\pi}$ for all $s$. Various entropic measures are possible, but the simplest is the negative log-likelihood: $\text{Entropy}(\bar{\lambda}^{\pi}) = -\sum_s \bar{\lambda}_s^{\pi} \log[\bar{\lambda}_s^{\pi}]$. As is well known, this entropy is (strongly) concave.

Another example, when $d$ state-action features $\phi(s, a) \in \mathbb{R}^d$ are available, is to cover the entire feature space by maximizing the smallest eigenvalue of the covariance matrix:

$$\max_{\pi} R(\pi) := \sigma_{\min}\left(\mathbb{E}^{\pi}\left[\sum_{t=1}^{\infty} \gamma^t \phi(s_t, a_t)\phi(s_t, a_t)^{\top}\right]\right). \qquad (8)$$

In (8), observe that $\mathbb{E}^{\pi}[\sum_{t=1}^{\infty} \gamma^t \phi(s_t, a_t)\phi(s_t, a_t)^{\top}] = \sum_{sa} \lambda_{sa}^{\pi} \cdot \phi(s, a)\phi(s, a)^{\top}$. By Rayleigh principle, it is again a concave function of $\lambda$.

**Example 2.3** (**Learning to mimic a demonstration**). When demonstrations are available, they may be employed to obtain information about a prior policy in the form of a state visitation distribution $\bar{\mu}$. Remaining close to this prior can be achieved by minimizing the Kullback-Liebler (KL) divergence between the state marginal distribution of $\lambda$ and the prior $\bar{\mu}$ stated as

$$F(\lambda) = \text{KL}\left((1 - \gamma)\sum_a \lambda_a || \bar{\mu}\right) \qquad (9)$$

which, when substituted into (4), yields a method for ensuring some baseline performance. We further note that in place of KL divergence, one can also use other convex distances such as Wasserstein, total variation, or Hellinger distances.

Additional instances may be found in [53]. With the setting clarified, we shift focus to developing an algorithmic solution to (4), that is, to solve for policy $\pi$.

## 3  Variational Policy Gradient Theorem

To handle the curse of dimensionality, we allow parametrization of the policy by $\pi = \pi_\theta$, where $\theta \in \Theta \subset \mathbb{R}^d$ is the parameter vector. In this way, we can narrow down the policy search problem to

within a $d$-dimensional parameter space rather than the high-dimensional space of tabular policies. The policy optimization problem then becomes

$$\max_{\theta \in \Theta} \ R(\pi_\theta) := F(\lambda^{\pi_\theta}) \tag{10}$$

where $F$ is the concave utility of the state-action occupancy measure $\lambda(\theta) := \lambda^{\pi_\theta}$, $\Theta \subset \mathbb{R}^d$ is a convex set. We seek to solve for the policy maximizing the utility as in (10) using gradient ascent over the parameter space $\Theta$. Note that (10) is simply (2) with parameterization $\theta$ of policy $\pi$ substituted. We denote by $\nabla_\theta R(\pi_\theta)$ the parameterized policy gradient of general utility.

First, recall the policy gradient theorem for RL with cumulative rewards [44]. Let the reward function be $r$. Define $V(\theta; r) := \langle \lambda(\theta), r \rangle$, i.e., the total expected discounted reward under the reward function $r$ and the policy $\pi_\theta$. The Policy Gradient Theorem states that

$$\nabla_\theta V(\theta; r) = \mathbb{E}^{\pi_\theta} \left[ \sum_{t=0}^{\infty} \gamma^t Q^{\pi_\theta}(s_t, a_t; r) \cdot \nabla_\theta \log \pi_\theta(a_t | s_t) \right], \tag{11}$$

where $Q^\pi(s, a; r) := \mathbb{E}^\pi \left[ \sum_t \gamma^t r(s_t, a_t) \mid s_0 = s, a_0 = a, a_t \sim \pi(\cdot \mid s_t) \right]$. Unfortunately, this elegant result no longer holds when we consider a general function instead of cumulative rewards: The policy gradient theorem relies on the additivity of rewards, which is lost in our problem. For future reference, we denote $Q^\pi(s, a; z) := \mathbb{E}^\pi \left[ \sum_t \gamma^t z_{s_t a_t} \mid s_0 = s, a_0 = a, a_t \sim \pi(\cdot \mid s_t) \right]$ where $z$ is any "function" of the state-action pairs ($z \in \mathbb{R}^{SA}$). Moreover, $V(\theta; z)$ is defined similarly. These definitions are motivated by subsequent efforts to derive an expression for the gradient of (10).

## 3.1 Policy Gradient of $R(\pi_\theta)$

Now we derive the policy gradient of $R(\pi_\theta)$ with respect to $\theta$. By the chain rule, the gradient of $F(\lambda(\theta)) := F(\lambda^{\pi_\theta})$, using the definition of $R(\pi_\theta)$, yields (assuming differentiability of $F, \lambda$):

$$\nabla_\theta R(\pi_\theta) = \sum_{s \in \mathcal{S}} \sum_{a \in \mathcal{A}} \frac{\partial F(\lambda(\theta))}{\partial \lambda_{sa}} \cdot \nabla_\theta \lambda_{sa}(\theta). \tag{12}$$

To directly use the chain rule, one needs the partial derivatives $\frac{\partial F(\lambda(\theta))}{\partial \lambda_{sa}}$ and $\nabla_\theta \lambda_{sa}(\theta)$. Unfortunately, neither of them is easy to estimate. The partial gradient $\frac{\partial F(\lambda(\theta))}{\partial \lambda_{sa}}$ is a function of the current state-action occupancy measure $\lambda^{\pi_\theta}$. One might attempt to estimate the measure $\lambda^{\pi_\theta}$ and then evaluate the gradient $\frac{\partial F(\lambda(\theta))}{\partial \lambda_{sa}}$. However, estimates of distributions over large spaces converge very slowly [46].

As it turns out, a viable alternate route is to consider the Fenchel dual $F^*$ of $F$. Recall that $F^*(z) = \inf_\lambda \langle \lambda, z \rangle - F(\lambda)$, where we use $\langle x, y \rangle := x^\top y$ (since $F$ is concave, the dual is defined using $\inf$, instead of $\sup$). As is well known, for $F$ concave, under mild regularity conditions, the bidual (dual of the dual) of $F$ is equal to $F$. This forms the basis of our first result, which states that the steepest policy ascent direction of (10) is the solution to a stochastic saddle point problem. The proofs of this and subsequent results are given in the supplementary material.

**Theorem 1** (**Variational Policy Gradient Theorem**). *Suppose $F$ is concave and continuously differentiable in an open neighborhood of $\lambda^{\pi_\theta}$. Denote $V(\theta; z)$ to be the cumulative value of policy $\pi_\theta$ when the reward function is $z$, and assume $\nabla_\theta V(\theta; z)$ always exists. Then we have*

$$\nabla_\theta R(\pi_\theta) = \lim_{\delta \to 0_+} \operatorname*{argmax}_x \inf_z \left\{ V(\theta; z) + \delta \nabla_\theta V(\theta; z)^\top x - F^*(z) - \frac{\delta}{2} \|x\|^2 \right\}. \tag{13}$$

Therefore, to estimate $\nabla_\theta R(\pi_\theta)$ we require the cumulative return $V(\theta; z)$ of the function $z$, its associated "vanilla" policy gradient (11), and the gradient of the Fenchel dual of $F$ at $z$. These ingredients are combined via (13) to obtain a valid policy gradient for general objectives. The entity $z$ we dub the "pseudo reward" as it emerges via the Fenchel conjugate in Theorem 1. This terminology is appropriate because $z$ plays the algorithmic role of a usual reward function although it is instead a statistic computed in terms of the reward. We also note that saddle-point reformulations of the classical policy gradient theorem have appeared recently [30], although Theorem 1 refers to more general utilities (2). Next, we discuss how to estimate the gradient using sampled trajectories.

## 3.2 Estimating the Variational Policy Gradient

Theorem 1 implies that one can estimate $\nabla_\theta R(\pi_\theta)$ by solving a stochastic saddle point problem. Suppose we generate $n$ i.i.d. episodes of length $K$ following $\pi_\theta$, denoted as $\zeta_i = \{s_k^{(i)}, a_k^{(i)}\}_{k=1}^K$. Then we can estimate $V(\theta; z)$ and $\nabla V(\theta; z)$ for any function $z$ by

$$\tilde{V}(\theta; z) := \frac{1}{n}\sum_{i=1}^n V(\theta; z; \zeta_i) := \frac{1}{n}\sum_{i=1}^n \sum_{k=1}^K \gamma^k \cdot z(s_k^{(i)}, a_k^{(i)}), \tag{14}$$

$$\nabla\tilde{V}(\theta; z) := \frac{1}{n}\sum_{i=1} \nabla_\theta V(\theta; z; \zeta_i) := \frac{1}{n}\sum_{i=1}^n \sum_{k=1}^K \sum_{a\in\mathcal{A}} \gamma^k \cdot Q(s_k^{(i)}, a; z)\nabla_\theta \pi_\theta(a|s_k^{(i)}).$$

For a given value of $K$, the error introduced by "truncating" trajectories at length $K$ is of order $\gamma^K/(1-\gamma)$, which quickly decays to zero for $\gamma < 1$. Plugging in the obtained estimates into (13) gives rise to the sample-average approximation to the policy gradient:

$$\hat{\nabla}_\theta R(\pi_\theta; \delta) := \underset{x}{\operatorname{argmax}} \inf_{\|z\|_\infty \leq \ell_F} \left\{ -F^*(z) + \tilde{V}(\theta; z) + \delta\nabla_\theta\tilde{V}(\theta; z)^\top x - \frac{\delta}{2}\|x\|^2 \right\}, \tag{15}$$

where $\ell_F$ is defined in the next theorem. Therefore, any algorithm that solves problem (15) will serve our purpose. A MC stochastic approximation scheme, i.e., Algorithm 1, is provided in Appendix B.1 of the supplementary material.

**Theorem 2 (Error bound of policy gradient estimates).** *Suppose the following holds:*
*(i) $dom F = \mathbb{R}^{SA}$, there exists $\ell_F$ such that $\max\{\|\nabla F(\lambda)\|_\infty : \|\lambda\|_1 \leq \frac{2}{1-\gamma}\} \leq \ell_F$.*
*(ii) $F$ is $L_F$-smooth under $L_1$ norm, i.e., $\|\nabla F(\lambda) - \nabla F(\lambda')\|_\infty \leq L_F\|\lambda - \lambda'\|_1$.*
*(iii) $F^*$ is $(\ell_{F^*})$-Lipschitz with respect to the $L_\infty$ norm in the set $\{z : \|z\|_\infty \leq 2\ell_F, F^*(z) > -\infty\}$.*
*(iv) There exists $C$ with $\|\nabla_\theta\pi(\cdot|s)\|_{\infty,2} \leq C$, where $\nabla_\theta\pi(\cdot|s) = [\nabla_\theta\pi(1|s), \cdots, \nabla_\theta\pi(A|s)]$.*
*Let $\hat{\nabla}_\theta R(\pi_\theta) := \lim_{\delta\to 0_+} \hat{\nabla}_\theta R(\pi_\theta; \delta)$. Then*

$$\mathbb{E}[\|\hat{\nabla}_\theta R(\pi_\theta) - \nabla_\theta R(\pi_\theta)\|^2] \leq \mathcal{O}\left(\frac{C^2(\ell_F^2 + L_F^2\ell_{F^*}^2)}{n(1-\gamma)^4} + \frac{C^2 L_F^2}{n(1-\gamma)^6}\right) + \mathcal{O}(\gamma^K).$$

**Remarks.**
**(1)** Theorem 2 suggests an $O(1/\sqrt{n})$ error rate, proving that the variational policy gradient - though more complicated than the typical policy gradient that takes the form of a mean - can be efficiently estimated from finite data.
**(2)** Although the variable $z$ is high dimensional, our error bound depends only on the properties of $F$.
**(3)** We assumed for simplicity that Q values are known. In practice, they can estimated by, e.g., an additional Monte Carlo rollout on the same sample path or temporal difference learning. As long as the estimator for $Q(s, a; z)$ is unbiased and upper bouded by $\mathcal{O}(\frac{\|z\|_\infty}{1-\gamma})$, the result will not change.
**(4)** For the case of cumulative rewards, we have $F(\lambda) = \langle r, \lambda \rangle$, so that $\ell_F = \|r\|_\infty$, $\ell_{F^*}=0$, $L_F=0$. Therefore $\mathbb{E}[\|\hat{\nabla}_\theta R(\pi_\theta) - \nabla_\theta R(\pi_\theta)\|^2] \leq \mathcal{O}\left(\frac{C^2\|r\|_\infty^2}{n(1-\gamma)^4}\right)$.

**Special cases of $\nabla_\theta R(\pi^\theta)$.** We further explain how to obtain the variational policy gradient for several special cases of $R$, including constrained MDP, maximal exploration, and learning from demonstrations. See Appendix B.2 in the supplementary for more details.

# 4 Global Convergence of Policy Gradient Ascent

In this section, we analyze policy search for the problem (10), i.e., $\max_{\theta\in\Theta} R(\pi_\theta)$ via gradient ascent:

$$\theta^{k+1} = \underset{\theta\in\Theta}{\operatorname{argmax}} R(\pi_{\theta^k}) + \langle\nabla_\theta R(\pi_{\theta^k}), \theta - \theta^k\rangle - \frac{1}{2\eta}\|\theta - \theta^k\|^2 = \operatorname{Proj}_\Theta\{\theta^k + \eta\nabla_\theta R(\pi_{\theta^k})\} \tag{16}$$

where $\operatorname{Proj}_\Theta\{\cdot\}$ denotes Euclidean projection onto $\Theta$, and equivalence holds by the convexity of $\Theta$.

### 4.1 No spurious first-order stationary solutions.

We study the geometry of the (possibly) nonconvex optimization problem (10). When $F$ is a linear function of $\lambda$, and the parametrization is tabular or softmax, existing theory of cumulative-return RL problems have shown that every first-order stationary point of (10) is globally optimal – see [2, 29].

In what follows, we show that the problem (10) has no spurious extrema despite of its nonconvexity, for general utility functions and policy parametrization. Specifically, to generalize global optimality attributes of stationary points of (10) from (1), we exploit structural aspects of the relationship between occupancy measures and parameterized families of policies, namely, that these entities are related through a bijection. This bijection, when combined with the fact that (10) is concave in $\lambda$, and suitably restricting the parameterized family of policies, is what we subsequently describe as "hidden convexity." For these results to be valid, we require the following regularity conditions.

**AS1.** *Suppose the following holds true:*
*(i). $\lambda(\cdot)$ forms a bijection between $\Theta$ and $\lambda(\Theta)$, where $\Theta$ and $\lambda(\Theta)$ are closed and convex.*
*(ii). The Jacobian matrix $\nabla_\theta \lambda(\theta)$ is Lipschitz continuous in $\Theta$.*
*(iii). Denote $g(\cdot) := \lambda^{-1}(\cdot)$ as the inverse mapping of $\lambda(\cdot)$. Then there exists $\ell_\theta > 0$ s.t. $\|g(\lambda) - g(\lambda')\| \leq \ell_\theta \|\|\lambda - \lambda'\|\|$ for some norm $\|\| \cdot \|\|$ and for all $\lambda, \lambda' \in \lambda(\Theta)$.*

In particular, for the direct policy parametrization, also known as the "tabular" policy case, we have $\lambda(\theta) := \Lambda(\pi)$ where $\Lambda$ is defined in (3). When $\xi$ is positive-valued, Assumption 1 is true for the tabular policy case (as established in Appendix H of the supplementary material). Note that the AS1 implicitly requires the minimum singular value of the Jacobian matrix $\nabla\lambda(\cdot)$ to be bounded away from 0 and the convex parameter set $\Theta$ to be compact. The result holds for tabular soft-max policy if $\Theta$ is restricted to a compact subset of the orthogonal complement of the all-one vectors, but it may not be valid for general softmax parametrization without additional regularization. It remains future work to understand the behavior of PG method under a broader family of policy parameterizations.

**Theorem 3** (**Global optimality of stationary policies**). *Suppose Assumption 1 holds, and $F$ is a concave, and continuous function defined in an open neighborhood containing $\lambda(\Theta)$. Let $\theta^*$ be a first-order stationary point of problem* (10)*, i.e.,*

$$\exists u^* \in \hat{\partial}(F \circ \lambda)(\theta^*), \quad s.t. \quad \langle u^*, \theta - \theta^* \rangle \leq 0 \qquad for \qquad \forall \theta \in \Theta. \tag{17}$$

*Then $\theta^*$ is a globally optimal solution of problem* (10)*.*

Theorem 3 provides conditions such that, despite of nonconvexity, local search methods can find the global optimal policies. Since we aim at general utilities, we naturally separated out the convex and non-convex maps in the composite objective and our conditions for optimality rely on the properties of these. In a recent paper, Bhandari and Russo [6] proposed some sufficient conditions under which a result similar to Theorem 3 holds in the setting of the standard, cumulative total reward criterion. Their conditions are *(i)* the policy class is closed under (one-step, weighted) policy improvement and that *(ii)* all stationary points of the one-step policy improvement map are global optima of this map. It remains for future work to see the relationship between our conditions and these conditions: They appear to have rather different natures.

### 4.2 Convergence analysis

Now we analyze the convergence rate of the policy gradient scheme (16) for general utilities.

**AS2.** *There exists $L > 0$ such that the policy gradient $\nabla_\theta R(\pi_\theta)$ is L-Lipschitz.*

The objective $R(\pi_\theta)$ is nonconvex in $\theta$, so one might expect that gradient schemes converge to stationary solutions at a standard $\mathcal{O}(1/\sqrt{t})$ convergence rate [42]. Remarkably, the policy optimization problem admits a convex nature if we view it in the space of $\lambda$, as long as $F$ is concave. By exploiting this hidden convexity, we establish an $\mathcal{O}(1/t)$ convergence rate for solving RL with general utilities. Further, we show that, when the utility $F$ is strongly concave, the gradient ascent scheme converges to the globally optimal policy exponentially fast.

**Theorem 4** (**Convergence rate of parameterized policy gradient iteration**). *Let Assumptions 1 and 2 hold. Denote $D_\lambda := \max_{\lambda, \lambda' \in \lambda(\Theta)} \|\|\lambda - \lambda'\|\|$ as defined in Assumption 1(iii). Then the policy gradient update* (16) *with $\eta = 1/L$ satisfies for all $k$*

$$R(\pi_{\theta^*}) - R(\pi_{\theta^k}) \leq \frac{4L\ell_\theta^2 D_\lambda^2}{k+1}.$$

*Additionally, if $F(\cdot)$ is $\mu$-strongly concave with respect to the $\|\!\|\cdot\|\!\|$ norm, we have*

$$R(\pi_{\theta^*}) - R(\pi_{\theta^k}) \leq \left(1 - \frac{1}{1 + L\ell_\theta^2/\mu}\right)^k (R(\pi_{\theta^*}) - R(\pi_{\theta^0})).$$

The exponential convergence result of Theorem 4 implies that, when a regularizer like Kullback-Leiber divergence is used, policy gradient method converges much faster. In other words, policy search with general utilities can actually be easier than the typical, cumulative-return problem.

Finally, we study the case where policies are not parameterized, i.e., $\theta = \pi$. The next theorem establishes a tighter convergence rate than what Theorem 4 already implies.

**Theorem 5** (**Convergence rate of tabular policy gradient iteration**). *Let $\theta = \pi$ and $\lambda(\theta) = \Lambda(\pi)$. Let Assumption 2 hold and assume that $\xi$ is positive-valued. Then the iterates generated by* (16) *with $\eta = 1/L$ satisfy for all $k \geq 1$ that*

$$R(\pi^*) - R(\pi^k) \leq \frac{20L|\mathcal{S}|}{(1-\gamma)^2(k+1)} \cdot \left\| d_\xi^{\pi^*}/\xi \right\|_\infty^2.$$

**The case of cumulative rewards.** Let us consider the well-studied special case where $F$ is a linear functional, i.e., $R(\pi) = V^\pi$ [cf. (1)] is the typical cumulative return. In this case, we have $L = \frac{2\gamma A}{(1-\gamma)^3}$ ([2]). Now in order to obtain an $\epsilon$-optimal policy $\bar{\pi}$ such that $V^{\pi^*} - V^{\bar{\pi}} \leq \epsilon$, the gradient ascent update requires $\mathcal{O}\left(\frac{SA}{(1-\gamma)^5\epsilon} \cdot \left\| d_\xi^\pi/\xi \right\|_\infty^2\right)$ iterations according to Theorem 5. This bound is strictly smaller than the $\mathcal{O}\left(\frac{SA}{(1-\gamma)^6\epsilon^2} \left\| d_\xi^{\pi^*}/\xi \right\|_\infty^2\right)$ iteration complexity proved by [2] for tabular policy gradient. The improvement from $O(1/\epsilon^2)$ to $(1/\epsilon)$ comes from the fact that, although the policy optimization problem is nonconvex, our analysis exploits its hidden convexity in the space of $\lambda$.

## 5 Experiments

Now we shift to numerically validating our methods and theory on OpenAI Frozen Lake [11]. Throughout, additional details may be found in Appendix C of the supplementary material..

**Policy Gradient (PG) Estimation.** First we investigate the use of Theorem 1 and Algorithm 1 (Appendix B.1 in the supplementary) for PG estimation, for several instances of the general utility. We also compare it with the gradient estimates computed by REINFORCE for cumulative returns. Specifically, in Figure 1 we illustrate the convergence of gradient estimates, measured using the

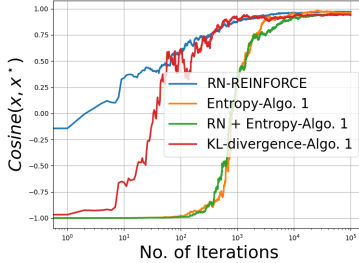

Figure 1: **PG estimation via Algorithm 1** Cosine similarity between PG estimates $\hat{x}_t$ generated by Algorithm 1 after $t$ samples and the ground truth $x^\star$, which consistently converges to near 1 across different instances (E.g. (2.1) - (2.3)) when $t$ becomes large. For comparison, we also include the convergence of PG estimates from REINFORCE for cumulative returns.

cosine similarity between $x_n$ (running estimate based on $n$ episodes) and the true gradient $x^*$ (which is evaluated using brute force Monte Carlo rollouts – see Appendix C.2 in the supplementary). The cosine similarity converges to 1 across different instances, providing evidence that Algorithm 1 yields consistent gradient estimates for general utilities.

**PG Ascent for Maximal Entropy Exploration.** Next, we consider maximum entropy exploration (7) using algorithm (16), with softmax parametrization. First, we display the evolution of the entropy

of the normalized occupancy measure over the number of episodes in Fig. 2(a). Then, we visualize the world model in Fig. 2(b)(bottom). Moreover, the lower middle is the occupancy measure associated with a uniformly random policy, the upper-middle layer visualizes the "pseudo-reward" $z^*$ computed as the Fenchel dual of the entropy (7) – see Appendix B.2 provided in the supplementary, which is null at the holes and positive otherwise. We use a different color to denote that its values are not likelihoods. The occupancy measure obtained by policy gradient ascent with gradient estimated by Algorithm 1 at the end of training is in Figure 2(b)(top) – observe the maximal entropy policy achieves significantly better coverage of the state space than the uniformly random policy.

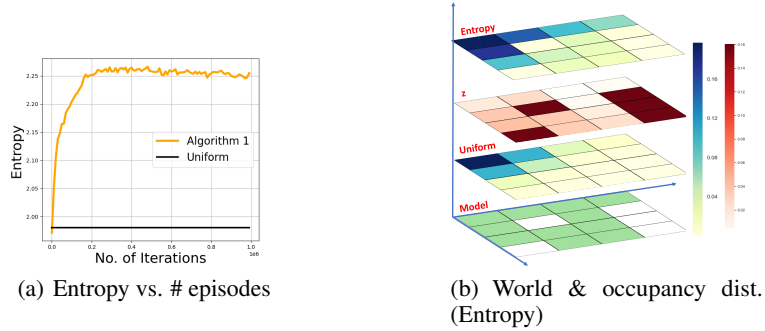

(a) Entropy vs. # episodes

(b) World & occupancy dist. (Entropy)

Figure 2: **Results for maximum entropy exploration**: In Fig. 2(a), to quantify exploration, we present the entropy of flux $\lambda$ over training index $n$ for our approach, as compared with the entropy of a uniform random policy. Fig. 2(b)(bottom) visualizes the world model (holes in the lake have null entropy, as they terminate the episode), the lower middle layer displays the occupancy measure associated with a uniformly random policy, the upper-middle visualizes the pseudo-reward $z^*$ defined by the Fenchel dual of the entropy (7) – see Appendix B.2 of the supplementary material. Lastly, on top we visualize the occupancy measure associated with the max entropy policy, which better covers the space than a uniformly random policy.

**PG Ascent for Avoiding Obstacles.** Suppose our goal is to navigate the Frozen Lake and avoid obstacles. We consider imposing penalties to avoid costly states [cf. (6)] via a logarithmic barrier (20), and by applying variational PG ascent, we obtain an optimal policy whose resulting occupancy measure is depicted in Fig. 3(a)(top). For comparison, we consider optimizing the standard expected cumulative return (1), whose state occupancy measure is given in Fig. 3(a)(middle). Observe that imposing log penalties yields policies whose probability mass is concentrated away from obstacles (dark green). Further, we display in Fig. 3 the reward 3(b) and cost 3(c) accumulation during test trajectories as a function of the iteration index for the PG ascent (16) for the cumulative return (1) as compared with a logarithmic barrier imposed to solve (6) for different penalty parameters $\beta$.

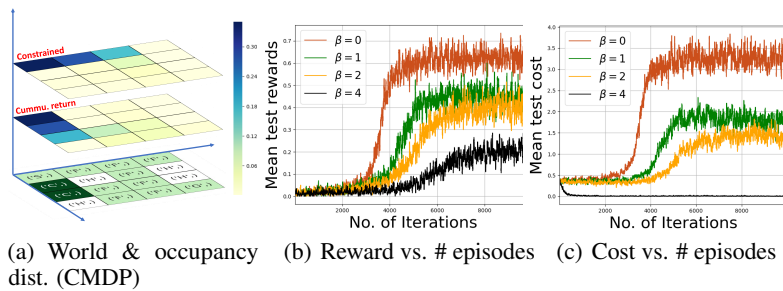

(a) World & occupancy dist. (CMDP)

(b) Reward vs. # episodes

(c) Cost vs. # episodes

Figure 3: **Results for avoiding obstacles.** Fig. 3(a)(bottom) depicts the world model of OpenAI Frozen Lake with augmentation to include costly states, e.g., obstacles: C represents costly states, F is the frozen lake, H is the hole, and G is the goal. We consider softmax policy parametrization, and visualize the occupancy measure associated with REINFORCE for the cumulative return (1) in the middle layer, and the relaxed **CMDP** (6) via a **logarithmic barrier** (20) at the top. The policy obtained via barriers avoids visiting costly states, in contrast to the middle. Fig. 3(b) and Fig. 3(c) show the reward/cost accumulated during test trajectories over training index for Algorithm 1. Observe that the reward/cost curves behave differently as the penalty parameter $\beta$ varies: observe that without any constraint imposition (which implies $\beta = 0$ in red), one achieves the highest reward, but incurs the most costs, i.e., hits obstacles most often. Larger $\beta$ imposes more penalty, and hence $\beta = 4$ incurs lowest cost and lowest reward. Other instances are also shown for $\beta = 1$ and $\beta = 2$.

# 6 Broader Impact

While RL has a great number of potential applications, our work is of foundational nature and as such, the application of the ideas in this paper can have both broad positive and negative impacts. However, this paper is purely theoretical, as we do not aim at any specific application, there is nothing we can say about the most likely broader impact of this work that would go beyond speculation.

## Acknowledgments and Disclosure of Funding

Mengdi Wang gratefully acknowledges funding from the U.S. National Science Foundation (NSF) grant CMMI1653435, Air Force Office of Scientific Research (AFOSR) grant FA9550-19-1-020, and C3.ai DTI.

Csaba Szepesvári gratefully acknowledges funding from the Canada CIFAR AI Chairs Program, Amii and NSERC.

Alec Koppel gratefully acknowledges funding from the SMART Scholarship for Service, ARL DSI-TRC Seedling, and the DCIST CRA.

## Footnotes

[1]We remark that the stationary policies are sufficient, because the set of occupancy measures generated by any policy is the same as that of generated by stationary policies [35, 18].

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
