[Supplementary Material]

# Supplementary Material for "Variational Policy Gradient Method for Reinforcement Learning with General Utilities"

## A   Related Work

We provide a more extension discussion for the context of this work. Firstly, when closed-form expressions for the optimizer of a function are unavailable, solving optimization problems requires iterative schemes such as gradient ascent [31]. Their convergence to global extrema is predicated on concavity and the tractability of computing ascent directions. When the objective takes the form of an expected value of a function parameterized by a random variable, stochastic approximations are required [36, 24]. The PG Theorem mentioned above gives a specific form for obtaining ascent directions with respect to a parameterized family of stationary policies via trajectories in a Markov decision process, when the objective is the expected cumulative return [44], which gives rise to the REINFORCE algorithm.

The convergence of policy search for the expected cumulative return has been studied extensively in recent years. Under general parameterizations the problem becomes nonconvex. Hence, early work focused on asymptotic convergence to stationarity [34] by invoking dynamical systems [8]. In actor-critic [26, 25], one replaces the Monte Carlo rollout of the Q function with a temporal difference estimator [43], and its asymptotic stability follows similar logic [7]. Another line of work focused on only on per-step value increase, i.e., policy improvement bounds [33, 34]. Recent interest has been on structural results that yield convergence to global optimality: when state transitions are linear [16, 12]), the policy parameterization is direct (tabular) [6, 2], function approximation error can be quantified [38, 27]. Clever step-size rules have also been designed to ensure convergence to second-order stationary points under general settings [55].

These results, however, are restricted to the expected cumulative return, a linear functional of the state-action occupancy measure, and hence do not apply to general concave functionals of the form considered in this work. Early works in operations research consider nonstandard utilities [19], motivated by certain variance-penalizations which may also be written as concave functionals of occupancy measures [17]. Similar in spirit to this work is [23], as it also puts occupancy measures at the center of its conceptual development. These works develop dynamic programming approaches for tabular settings, and hence are not scalable to problems with large spaces. More recently, maximizing the entropy of the state visitation distribution has been considered [18], a special case of the concave utilities we study. Moreover, the authors develop a model-based iteratively policy update, which requires explicit knowledge of the transition probability matrix. By contrast, in this work we prioritize model-free approaches for possibly large spaces via the fusion of direct policy search and parameterization over a family of policies. Related notions of hidden convexity have been studied in related maximum likelihood scenarios over non-concave distributions recently [13].

## B   Supplementary materials of Section 3

### B.1   A Monte Carlo Algorithm for solving (15)

Note that any algorithm that solves problem (15) will serve our purpose. Therefore, we provide a Monte Carlo method that alternates between stochastic primal and dual updates as an example, stated in Algorithm 1, in which the projection operator onto the set $\{z : \|z\|_\infty \leq \ell_F\}$ is denoted as $\mathrm{Proj}_{\ell_F}\{z\}$. For any $z$, $z' = \mathrm{Proj}_{\ell_F}\{z\}$ is defined as

$$z'_i = \begin{cases} -\ell_F, & \text{if } z_i \in (-\infty, -\ell_F), \\ z_i, & \text{if } z_i \in [-\ell_F, \ell_F], \\ \ell_F, & \text{if } z_i \in (\ell_F, +\infty). \end{cases}$$

It is worth noting that we omit the term $\delta \nabla \tilde{V}(\theta; z)^\top x$ when computing the gradient w.r.t. $z$ in (18). Note that for the iterates $x^t$ are all well bounded, then $\delta \nabla \tilde{V}(\theta; z^t)^\top x^t = \mathcal{O}(\delta)$, which is negligible when $\delta \to 0$.

**Algorithm 1** Monte Carlo Variational Policy Gradient Estimation

---

**Require:** a differentiable policy parametrization $\pi_\theta$, stepsizes $\alpha_t, \beta_t > 0$, initial points $\boldsymbol{x} = 0$, $\mathbf{z} = 0$.
A constant $\ell_F$.

**policy parameter** $\boldsymbol{\theta} \in \mathbb{R}^d$

Generate episodes $\zeta_i = \{(s_{k_t}, a_{k_t})\}$ from $i = 1, \cdots, n$ following $\pi_\theta(a|s)$

For $t = 0, 1, 2, \ldots$ until some stopping criterion is met:

    **Sample** $(s_{k_t}, a_{k_t})$ from the data set

    **Update**

$$\mathbf{z}^{t+1} \leftarrow \operatorname{Proj}_{\ell_F} \left\{ \mathbf{z}^t - \frac{\alpha_t}{1-\gamma} \mathbf{1}_{s_{k_t}, a_{k_t}} + \alpha_t \nabla F^*(\mathbf{z}^t) \right\} \qquad (18)$$

$$\boldsymbol{x}^{t+1} \leftarrow \boldsymbol{x}^t + \beta_t \left[ \sum_{a \in \mathcal{A}} Q^{\pi_\theta}(s_{k_t}, a; \mathbf{z}^t) \cdot \nabla_\theta \pi_\theta(a|s_{k_t}) - \boldsymbol{x}^t \right] \qquad (19)$$

**Output:** the last iterate $\boldsymbol{x}$

---

## B.2   Special cases of policy gradient computation

We give several examples of the policy gradient for special cases of the general utility in (10).

**Linear utility**   The simplest, where $F(\lambda) = \langle \lambda, r \rangle$ [cf. (1)], we have $F^*(z) = 0$ if $z = c \cdot r$ for some scalar $c$ and $F^*(z) = \infty$ otherwise. In this case $z^* = r$ and Theorem 1 recovers the known policy gradient theorem for the risk-neutral MDP (1), that is $\nabla_\theta R(\pi_\theta) = \nabla_\theta V(\theta; r)$.

**Constrained MDPs**   By contrast, in Example 2.1, i.e., when a constraint $\mathbb{E}^\pi \left[ \sum_{t=0}^\infty \gamma^t c(s_t, a_t) \right] \leq C$ on the accumulation of costs $c(s_t, a_t)$ is present, and we may enforce it approximately with a log barrier by defining

$$R(\pi_\theta) = \langle r, \lambda(\theta) \rangle + \beta \log\left(C - \langle c, \lambda(\theta) \rangle\right) = V(\theta; r) + \beta \log\left(C - V(\theta; c)\right), \qquad (20)$$

where $\beta$ is a regularization parameter, in which case the policy gradient takes the form

$$\nabla R(\pi_\theta) = \nabla_\theta V(\theta; r) - \beta \frac{\nabla_\theta V(\theta; c)}{C - V(\theta; c)}.$$

Estimating the policy gradient $R$ of constrained MDP consists of estimating two policy gradients $\nabla_\theta V(\theta; c)$ and $\nabla_\theta V(\theta; r)$ and accumulated reward $V(\theta; c)$.

**Minimum eigenvalue**   For case (8), define $\Phi(\lambda^{\pi_\theta}) = \sum_{s,a} \lambda_{sa}^{\pi_\theta} \cdot \phi(s, a) \phi(s, a)^\top$. Then $\Phi(\lambda^{\pi_\theta})$ is symmetric and positive semidefinite, since $\lambda^{\pi_\theta} \geq 0$. By using Rayleigh principle, we have

$$R(\pi_\theta) = \sigma_{min}\left(\Phi(\lambda^{\pi_\theta})\right) = \min_{\|u\|=1} u^\top \Phi(\lambda^{\pi_\theta}) u = \min_{\|u\|=1} \sum_{s,a} \lambda_{sa}^{\pi_\theta} |\phi(s, a)^\top u|^2. \qquad (21)$$

which is the minimum of a family of linear function in $\lambda$. Let $v^{(1)}, ..., v^{(k)}$ be a group of orthonormal bases of the eigenspace of $\Phi(\lambda^{\pi_\theta})$ corresponding to the minimum eigenvalue. Then define $k$ vectors as $r^{(i)}(s, a) = |\phi(s, a)^\top v^{(i)}|^2, \forall s, a, i = 1, ..., k$. Then the Fréchet superdifferential of $R$ at $\theta$ is

$$\hat{\partial}_\theta R(\pi_\theta) = \left\{ \nabla_\theta V(\theta; r) : r \in \operatorname{conv}(r^{(1)}, ..., r^{(k)}) \right\},$$

where $\operatorname{conv}(\cdot)$ denotes the convex hull of a group of vectors. When the multiplicity of the minimum eigenvalue is 1, then $R(\cdot)$ is differentiable at this point and $\hat{\partial}_\theta R(\cdot) = \{\nabla_\theta R(\cdot)\}$.

**Entropy maximization**   For the entropy (7), its Fenchel dual takes the form

$$F^*(z) = -\sum_{sa} \exp\left\{ -\frac{z_{sa}}{1-\gamma} - 1 \right\}.$$

**Learning to mimic a distribution**   For the KL divergence to a prior $\mu$ in (9), we have

$$F^*(z) = \begin{cases} -\sum_s \mu_s \exp\left\{ -\frac{z_{s1}}{1-\gamma} - 1 \right\} & \text{if} \quad z_{sa_1} = z_{sa_2} \quad \forall s \in \mathcal{S}, a_1, a_2 \in \mathcal{A}, \\ -\infty & \text{otherwise.} \end{cases}$$

## C   Additional Details of Experiments

### C.1   Details of Environment

OpenAI Frozen Lake is a finite-state action problem. The standard state consists of $\{S, F, H, G\}$, to which we add an additional state $C$ which is visualized in Fig. 3(a). At each step, an agent selects an action $a \in \mathcal{A}$, which consists of one of four directions (up, down, left, right), which may be enumerated as $\{1, \ldots, 4\}$. The reward is null at all Frozen $F$ spaces, the start $S$ location, and the Holes $H$ in the lake. If the agent enters a hole, the episode terminates, and hence null reward is accumulated for this trajectory. The only positive reward is $1$ and may be obtained when reaching the goal state $G$. Our augmentation is that costly states $C$ have been added, which incur reward $-0.4$ to represent, for instance, obstacles. We note that only for the cumulative return and its constrained variants, or other utilities that are defined in terms of the problem's inherent reward do these quantities matter. That is, for the entropy maximization problem, there is no reward associated with any state in the usual sense. The MDP transition model is unknown and defined by the OpenAI environment, a simulation oracle that provides state-action-reward triples.

Throughout all experiments, for simplicity, we considered a softmax policy parameterization. For this parameterization, the policy takes the form $\pi_\theta(s \mid a) = e^{\theta_{sa}} / (\sum_{a'} e^{\theta_{sa'}})$ for $\theta \in \mathbb{R}^{|\mathcal{S}| \times |\mathcal{A}|}$. For the Frozen lake environment in this paper, we have $|\mathcal{S}| = 16$ and $|\mathcal{A}| = 4$.

### C.2   Computing the True Policy Gradient

For comparison, we compute the true policy gradient by using a baseline approach based on the chain rule and a variant of REINFORCE [44]: the second factor on the right-hand side of (12) is exactly computed using REINFORCE $\nabla_\theta \lambda_{sa}(\theta)$, whereas the first, $\frac{\partial F(\lambda(\theta))}{\partial \lambda_{sa}}$, is computed using an additional Monte Carlo rollout. We denote as $x^*$ the result of this procedure and use it as ground truth. In Figure 4(a) we display the evolution of its norm difference $\|\hat{x}_n^\star - \hat{x}_{n-1}^\star\|$ as the sample size $n$ increases. That it approaches null with the sample size implies that this brute force Monte Carlo variant of REINFORCE is convergent, and hence is a reasonable benchmark comparator.

(a) Convergence of $x^\star$

Figure 4: Fig. 4(a) displays the convergence of a generalization of REINFORCE-based gradient estimator for (12) in terms of its difference $\|\hat{x}_n^\star - \hat{x}_{n-1}^\star\|$ as the number of processed trajectories $n$ increases, which converges to null, certifying $\hat{x}_n^\star$ as a baseline.

### C.3   Details about Maximum Entropy Exploration

For this problem instance, i.e., (7) from Example 2.2, we also consider the state space defined by Frozen Lake, but note that the reward as defined by the environment is now a moot point. This is

because each state contributes positive entropy, with the exception of the holes in the lake, which terminate the episode. We visualize this setup at the bottom layer of Fig. 2(b). The lower middle layer visualizes the occupancy measure associated with a uniform policy. Moreover, the upper middle layer visualizes the "pseudo-reward" $z$ for each point in the state space. This quantity is computed in terms of the Fenchel dual of the entropy – see Appendix B.2, and the occupancy measure associated with the output of Algorithm 1 at the end of training is visualized at the top layer. To obtain this result, we run it for $10^5$ total episodes, and for each episode we evaluate the entropy using (7). We consider a constant step-size $\alpha = 0.01, \beta = 0.1$, and $\eta = 0.001$ throughout this experiment.

### C.4 Details about the Constrained Markov Decision Process

In this subsection, we elaborate upon the implementation of Example 2.1, specifically, (6) and its approximation using a logarithmic barrier as detailed in (20). We consider the problem of navigating through the FrozenLake environment as shown in Fig. 3(a)(bottom): we seek to reach the goal state $G$ (reward = 1) from the starting location $S$ (reward = 0), navigating along $F$ frozen spaces (reward = 0), while avoiding locations marked $C$ (reward = $-0.2$) that denote costly states (obstacles) and $H$ holes.

We consider two approaches to the problem: first, we focus on optimizing the standard expected cumulative return (1), whose associated state occupancy measure is given in Fig. 3(a)(middle); second, we consider imposing constraints to avoid costly states [cf. (6)] via a logarithmic barrier (20), whose resulting occupancy measure is depicted in Fig. 3(a)(top). Bluer/yellower colors denote higher/lower likelihoods, respectively. We observe that imposing constraints yields policies whose probability mass is concentrated away from constraints and instead along paths from the start to the goal. Thus, Algorithm 1 combined with a policy search scheme (16) may be used to solve CMDPs.

This trend is corroborated in Fig. 3, which depicts the reward 3(b) and cost 3(c) accumulation during test trajectories as a function of training index for Algorithm 1 for the cumulative return (1) as compared with a logarithmic barrier imposed to solve CMDP (6) for different penalty parameters $\beta$. We may observe that without imposing any constraint ($\beta = 0$ in red), one achieves the highest reward, but incurs the most costs, i.e., hits obstacles most often, a form of "reckless boldness." Larger $\beta$ means higher penalty for the constraints, and hence $\beta = 4$ incurs lower cost and lower reward. We further added the curves for $\beta = 1$ and $\beta = 2$ for comparison.

For all results reported in Fig. 3, we run the algorithm for $10K$ total training steps in the form of episodes. For each episode, we run a number of evaluation (test) trajectories in order to determine their merit, both in terms of reward and cost accumulation. Put more simply, we evaluate the performance averaged over a few test trajectories as a function of episode number and report its average over last 20 episodes to show the trend. This is to illuminate policy improvement in its various forms (reward/cost accumulation) during training. Moreover, the algorithm is run with constant step-size $\eta = 0.1$ throughout this experiment.

## D  Proof of Theorem 1

*Proof.* First note that for any $z \in \mathbb{R}^{SA}, x \in \mathbb{R}^d$, we have

$$V(\theta; z) = \langle z, \lambda(\theta) \rangle,$$
$$\nabla_\theta V(\theta; z)^\top x = \langle z, \nabla_\theta \lambda(\theta) x \rangle, \tag{22}$$

where $\nabla_\theta \lambda(\theta)$ is the $SA \times d$ Jacobian matrix, the first identity holds by definition, and the second holds by directly differentiating the first identity and product it with $x$.

Consider the saddle point problem in (13) for fixed $0 < \delta < 1$. Let $G$ be any constant such that $\|\nabla F(\lambda(\theta))\|_\infty < G$. Define

$$(x^*(\delta), z^*(\delta)) \quad := \operatorname{argmax}_x \operatorname{argmin}_{\|z\|_\infty \leq G} \left\{ V(\theta; z) + \delta \nabla_\theta V(\theta; z)^\top x - F^*(z) - \tfrac{\delta}{2} \|x\|^2 \right\} \tag{23}$$

Note in (23) we added the auxiliary constraint set $\{z : \|z\|_\infty \leq G\}$, and later we will show that this constraint is inactive for all $\delta$ sufficiently small. We will also show that $(x^*(\delta), z^*(\delta))$ are bounded for all $\delta$ sufficiently small.

By the first-order stationarity condition, we have

$$x^*(\delta) = \nabla_\theta V(\theta; z^*(\delta)).$$

Note that $\nabla_\theta V(\theta; \cdot)$ is a linear function of $z$, thus there exists $B > 0$ such that $\|\nabla_\theta V(\theta; z)\| \le B$ for all $z \in \{\|z\|_\infty \le G\}$. And consequently $\|x^*(\delta)\| \le B$ for all $\delta > 0$.

For all $x \in \{\|x\| \le 2B\}$, we have

$$\lim_{\delta \to 0_+} \lambda(\theta) + \delta \nabla_\theta \lambda(\theta) x = \lambda(\theta).$$

Therefore, there exists some small $\delta_0 > 0$, such that for all $\delta < \delta_0$, the vector $\lambda(\theta) + \delta \nabla_\theta \lambda(\theta) x$ belongs to the neighborhood on which $F$ is differentiable and

$$\|\nabla F(\lambda(\theta) + \delta \nabla_\theta \lambda(\theta) x)\|_\infty < G, \quad \forall \ x \in \{x : \|x\| \le 2B\}.$$

In this case, we consider the unconstrained solution, for $\|x\| \le 2B$, defined by

$$z^*(x; \delta) := \operatorname*{argmin}_z V(\theta; z) + \delta \nabla_\theta V(\theta; z)^\top x - F^*(z) = \nabla F(\lambda(\theta) + \delta \nabla_\theta \lambda(\theta) x),$$

and observe that the unconstrained solution satisfies $\|z^*(x; \delta)\|_\infty < G$, and consequently the constraint $\|z\|_\infty \le G$ is not active. Therefore, for $\delta < \delta_0$, we can equivalently rewrite (23) as

$$
\begin{aligned}
x^*(\delta) &:= \operatorname*{argmax}_{\|x\| \le 2B} \min_z \left\{ V(\theta; z) + \delta \nabla_\theta V(\theta; z)^\top x - F^*(z) - \frac{\delta}{2} \|x\|^2 \right\} \quad (24)\\
&= \operatorname*{argmax}_{\|x\| \le 2B} F(\lambda(\theta) + \delta \nabla_\theta \lambda(\theta) x) - \frac{\delta}{2} \|x\|^2,
\end{aligned}
$$

Recall that we showed $\|x^*(\delta)\| \le B$, therefore the constraint $\|x\| \le 2B$ is also inactive and removable. Therefore $x^*(\delta)$ is equivalent to the unconstrained min-max solution, for all $\delta$ sufficiently small, and Fenchel duality together with the first-order stationarity condition implies

$$
\begin{aligned}
x^*(\delta) &= \operatorname*{argmax}_x \inf_z \left\{ V(\theta; z) + \delta \nabla_\theta V(\theta; z)^\top x - F^*(z) - \frac{\delta}{2} \|x\|^2 \right\}\\
&= \nabla_\theta \lambda(\theta)^\top \nabla F(\lambda(\theta) + \delta \nabla_\theta \lambda(\theta) x^*(\delta)).
\end{aligned}
$$

By using the fact that $\nabla F$ is continuous at $\lambda(\theta)$ and $x^*(\delta)$ is bounded, by letting $\delta \to 0$ on both sides, we get

$$
\begin{aligned}
\lim_{\delta \to 0_+} x^*(\delta) &= \lim_{\delta \to 0_+} \nabla_\theta \lambda(\theta)^\top \nabla F(\lambda(\theta) + \delta \nabla_\theta \lambda(\theta) x^*(\delta))\\
&= \nabla_\theta \lambda(\theta)^\top \nabla F(\lambda(\theta))\\
&= \nabla R(\theta),
\end{aligned}
$$

where the last equality uses the chain rule.

$\square$

# E    Proof of Theorem 2

*Proof.* First, let us denote the expression in (13) for fixed $0 < \delta < 1$ as

$$(x^*(\delta), z^*(\delta)) = \operatorname*{argmax}_x \operatorname*{argmin}_{\|z\|_\infty \le \ell_F} V(\theta; z) + \delta \nabla_\theta V(\theta; z)^\top x - F^*(z) - \frac{\delta}{2} \|x\|^2, \quad (25)$$

and its approximation with empirically estimated value functions and their gradients in (14) as

$$(\hat{x}(\delta), \hat{z}(\delta)) = \operatorname*{argmax}_x \operatorname*{argmin}_{\|z\|_\infty \le \ell_F} \tilde{V}(\theta; z) + \delta \nabla_\theta \tilde{V}(\theta; z)^\top x - F^*(z) - \frac{\delta}{2} \|x\|^2. \quad (26)$$

Then we decompose the entity $\mathbb{E}\left[ \left\| \hat{\nabla}_\theta R(\pi_\theta) - \nabla_\theta R(\pi_\theta) \right\|^2 \right]$ into three terms by adding and subtracting (i) $x^*(\delta)$ and (ii) $\hat{x}(\delta)$, which we then establish depends on the difference between (iii) $\hat{z}(\delta)$ and $z^*(\delta)$. Taken together with computing the limit of the right-hand side as $\delta \to 0$ we obtain the result. Each of these steps is analyzed independently, whose estimation errors are derived in the following lemma.

**Lemma 1.** *Consider $(x^*(\delta), z^*(\delta))$ and $(\hat{x}(\delta), \hat{z}(\delta))$ as defined in (25)-(26), respectively. Under the technical conditions stated in Theorem 2, their respective estimation errors satisfy:*

*(i)* $\left\| x^*(\delta) - \nabla_\theta R(\pi_\theta) \right\|^2 = \mathcal{O}(\delta^2).$

*(ii)* $\mathbb{E}\left[ \left\| x^*(\delta) - \hat{x}(\delta) \right\|^2 \right] \leq \frac{2C^2 \|z^*(\delta)\|_\infty^2}{(1-\gamma)^4} \cdot \left( \frac{\gamma^{2K}}{(1-\gamma)^2} + \frac{1}{n} \right) + \frac{2C^2}{(1-\gamma)^4} \cdot \mathbb{E}\left[ \|z^*(\delta) - \hat{z}(\delta)\|_\infty^2 \right].$

*(iii)* $\mathbb{E}\left[ \|\hat{z}(\delta) - z^*(\delta)\|_\infty^2 \right] \leq \mathcal{O}\left( \frac{L_F^2}{n(1-\gamma)^2} + \frac{L_F^2 \ell_{F*}^2}{n} + \frac{L_F^2 \delta^2 + L_F \delta}{n} \right).$

Combining the three steps and the fact that $\|z^*(\delta)\|_\infty \leq \ell_F$ yields

$$\mathbb{E}\left[ \|\hat{x}(\delta) - \nabla_\theta R(\theta)\|^2 \right] \leq \mathcal{O}\left( \frac{C^2(\ell_F^2 + L_F^2 \ell_{F*}^2)}{n(1-\gamma)^4} + \frac{C^2 L_F^2}{n(1-\gamma)^6} \right) + \mathcal{O}(\delta^2 + \delta/n + \gamma^K).$$

Let $\delta \to 0$, we get

$$\mathbb{E}\left[ \left\| \hat{\nabla}_\theta R(\pi_\theta) - \nabla_\theta R(\pi_\theta) \right\|^2 \right] \leq \mathcal{O}\left( \frac{C^2(\ell_F^2 + L_F^2 \ell_{F*}^2)}{n(1-\gamma)^4} + \frac{C^2 L_F^2}{n(1-\gamma)^6} \right) + \mathcal{O}(\gamma^K).$$

Lemma 1(i) - (iii) is proved in the next subsection. For the ease of notation, we will simply denote $x^*$ and $\hat{x}$ instead of $x^*(\delta)$ and $\hat{x}(\delta)$. Similarly, we denote $z^*$ and $\hat{z}$ instead of $z^*(\delta)$ and $\hat{z}(\delta)$.

$\square$

## E.1 Preliminary Technicalities

**Linearity property**. The functions $Q$, $V$ and $\nabla_\theta V$ are linear in the reward function. Namely, for any $\alpha, \alpha' \in \mathbb{R}$ and $r, r' \in \mathbb{R}^{|\mathcal{S}||\mathcal{A}|}$,

$$\alpha \nabla_\theta V(\theta; r) + \alpha' \nabla_\theta V(\theta; r') = \nabla_\theta V(\theta; \alpha r + \alpha' r').$$

Similar identities holds for $Q^{\pi_\theta}(s, a; \cdot)$ and $V(\theta; \cdot)$. For the stochastic estimators $\nabla_\theta \tilde{V}(\theta; r; \zeta)$, it is straightforward to check that the linearity property is still true.
**Upperbounding $Q$ and $V$.** Given an arbitrary reward function $r$, the upper bounds of $Q$ and $V$ functions are

$$|Q^{\pi_\theta}(s, a; r)| \leq \frac{\|r\|_\infty}{1-\gamma} \qquad \text{and} \qquad |V(\theta; r)| \leq \frac{\|r\|_\infty}{1-\gamma}.$$

**Uniform upperbounds for estimators.** Given any sample path $\zeta = \{(s_k, a_k)\}_{k=0}^K$, the estimators $\tilde{V}(\theta; z; \zeta)$ and $\nabla_\theta \tilde{V}(\theta; z; \zeta)$ are upper bounded by

$$\tilde{V}(\theta; z; \zeta)| \leq \frac{\|z\|_\infty}{1-\gamma} \qquad \text{and} \qquad \|\nabla_\theta \tilde{V}(\theta; z; \zeta)\| \leq \frac{C\|z\|_\infty}{(1-\gamma)^2}. \tag{27}$$

Consequently, as the sample averages of $\tilde{V}(\theta; z; \zeta_i)$ and $\nabla_\theta \tilde{V}(\theta; z; \zeta_i)$, we also have

$$|\tilde{V}(\theta; z)| \leq \frac{\|z\|_\infty}{1-\gamma} \qquad \text{and} \qquad \|\nabla_\theta \tilde{V}(\theta; z)\| \leq \frac{C\|z\|_\infty}{(1-\gamma)^2} \tag{28}$$

for any set of sample paths $\{\zeta_i\}_{i=1}^n$.

*Proof.* For $\tilde{V}(\theta; z; \zeta)$, for any $z$,

$$|\tilde{V}(\theta; z; \zeta)| = \left| \sum_{k=0}^K \gamma^k \cdot z(s_k, a_k) \right| \leq \sum_{k=0}^K \gamma^k \|z\|_\infty \leq \frac{\|z\|_\infty}{1-\gamma}$$

For $\nabla_\theta \tilde{V}(\theta; z; \zeta)$, for any $z$,

$$
\begin{aligned}
\|\nabla_\theta \tilde{V}(\theta; z; \zeta)\| &= \left\| \sum_{k=1}^{K} \sum_{a \in \mathcal{A}} \gamma^k \cdot Q(s_k, a; z) \nabla_\theta \pi_\theta(a|s_k) \right\| \\
&\leq \sum_{k=1}^{K} \gamma^k \cdot \left\| \sum_{a \in \mathcal{A}} Q(s_k, a; z) \nabla_\theta \pi_\theta(a|s_k) \right\| \\
&\leq \sum_{k=1}^{K} \gamma^k \cdot \max_{\|u\|_\infty \leq \frac{\|z\|_\infty}{1-\gamma}} \|\pi_\theta(\cdot|s_k)u\| \\
&\leq \frac{C\|z\|_\infty}{(1-\gamma)^2}.
\end{aligned}
$$

$\square$

## E.2 Proof of Lemma 1(i).

Consider the problem (25). First let us ignore the requirement that $\|z\|_\infty \leq \ell_F$. For this series of unconstrained problem, Theorem 1 suggests that

$$
\lim_{\delta \to 0_+} x^*(\delta) = \nabla_\theta R(\pi_\theta).
$$

Consequently, $\lim_{\delta \to 0_+} \lambda(\theta) + \delta \nabla_\theta \lambda(\theta) x^*(\delta) = \lambda(\theta)$. Because $\|\lambda(\theta)\|_1 = (1-\gamma)^{-1}$, $\exists \delta_0 > 0$ s.t. when $\delta < \delta_0$ we have

$$
\|\lambda(\theta) + \delta \nabla_\theta \lambda(\theta) x^*(\delta)\|_1 \leq \frac{2}{1-\gamma}.
$$

According to condition (i) of this theorem, we have

$$
\|\nabla F(\lambda(\theta) + \delta \nabla_\theta \lambda(\theta) x^*(\delta))\|_\infty \leq \ell_F.
$$

It is worth noting that $z^*(\delta) = \nabla F(\lambda(\theta) + \delta \nabla_\theta \lambda(\theta) x^*(\delta))$ is also the solution to the unconstrained version of (25). Therefore we have $\|z\|_\infty \leq \ell_F$, so that we can add this to the constraint without changing the optimal solutions. By the intermediate result in the proof of Theorem 1, we have

$$
x^*(\delta) = \nabla_\theta \lambda(\theta)^\top \nabla F \big( \lambda(\theta) + \delta \nabla_\theta \lambda(\theta) x^*(\delta) \big).
$$

Consequently, by the Lipschitz continuity of $\nabla F$, we have

$$
\begin{aligned}
\left\| x^*(\delta) - \nabla_\theta R(\theta) \right\|^2 &= \left\| \nabla_\theta \lambda(\theta)^\top \nabla F \big( \lambda(\theta) + \delta \nabla_\theta \lambda(\theta) x^*(\delta) \big) - \nabla_\theta \lambda(\theta)^\top \nabla F \big( \lambda(\theta) \big) \right\|^2 \\
&\leq \|\nabla_\theta \lambda(\theta)^\top\|_{\infty,2} \cdot \left\| \nabla F \big( \lambda(\theta) + \delta \nabla_\theta \lambda(\theta) x^*(\delta) \big) - \nabla F \big( \lambda(\theta) \big) \right\|_\infty^2 \\
&\leq L_F \|\nabla_\theta \lambda(\theta)^\top\|_{\infty,2}^2 \cdot \left\| \delta \nabla_\theta \lambda(\theta) x^*(\delta) \right\|_1^2 \\
&= \mathcal{O}(\delta^2).
\end{aligned}
$$

as stated in Lemma 1(i). In the last step, we used the fact that $x^*(\delta)$ is bounded because $x^*(\delta) \to \nabla_\theta R(\pi_\theta)$. $\square$

## E.3 Proof of Lemma 1(ii).

By the first order stationarity condition of the problems (25)-(26), we know

$$
x^* = \nabla_\theta V(\theta; z^*) \qquad \text{and} \qquad \hat{x} = \nabla_\theta \tilde{V}(\theta; \hat{z}).
$$

Consider the norm-difference between the preceding quantities:

$$
\mathbb{E}\left[ \|x^* - \hat{x}\|^2 \right] \leq 2\mathbb{E}\left[ \|\nabla_\theta V(\theta; z^*) - \nabla_\theta \tilde{V}(\theta; z^*)\|^2 \right] + 2\mathbb{E}\left[ \|\nabla_\theta \tilde{V}(\theta; z^*) - \nabla_\theta \tilde{V}(\theta; \hat{z})\|^2 \right]. \quad (29)
$$

To bound the term $\mathbb{E}\left[\|\nabla_\theta V(\theta; z^*) - \nabla_\theta \tilde{V}(\theta; z^*)\|^2\right]$, recall the definition (14):

$$\nabla \tilde{V}(\theta; z) := \frac{1}{n} \sum_{i=1}^n \nabla_\theta V(\theta; z; \zeta_i) = \frac{1}{n} \sum_{i=1}^n \sum_{k=1}^K \sum_{a \in \mathcal{A}} \gamma^k Q(s_k^{(i)}, a; z) \nabla_\theta \pi_\theta(a|s_k^{(i)}).$$

Consider the first term on the right-hand side of (29). Add and subtract $\mathbb{E}\left[\nabla_\theta \tilde{V}(\theta; z^*)\right]$ and use the fact that $\mathbb{E}\left[\nabla_\theta \tilde{V}(\theta; z^*)\right] = \nabla_\theta V(\theta; z^*)$, i.e., the bias-variance decomposition identity, to write

$$\mathbb{E}\left[\left\|\nabla_\theta V(\theta; z^*) - \nabla_\theta \tilde{V}(\theta; z^*)\right\|^2\right] \tag{30}$$
$$= \left\|\nabla_\theta V(\theta; z^*) - \mathbb{E}\left[\nabla_\theta \tilde{V}(\theta; z^*)\right]\right\|^2 + \mathbb{E}\left[\left\|\nabla_\theta \tilde{V}(\theta; z^*) - \mathbb{E}\left[\nabla_\theta \tilde{V}(\theta; z^*)\right]\right\|^2\right].$$

For the first (squared bias) term on the right-hand side of (30), denote $d_{\xi,K}^\pi(s) = (1-\gamma) \sum_{t=0}^K \gamma^t \mathbf{Prob}(s_t = s | \pi, s_0 \sim \xi)$. Then it is straightforward that $\sum_s |d_{\xi,K}^\pi(s) - d_\xi^\pi(s)| \le \frac{\gamma^K}{1-\gamma}$. As a result, we know

$$\left\|\nabla_\theta V(\theta; z^*) - \mathbb{E}\left[\nabla_\theta \tilde{V}(\theta; z^*)\right]\right\|^2 \tag{31}$$
$$= \frac{1}{(1-\gamma)^2} \left\|\sum_s \left(d_\xi^\pi(s) - d_{\xi,K}^\pi(s)\right) \sum_a Q^{\pi_\theta}(s, a; z^*) \nabla_\theta \pi_\theta(a|s)\right\|^2$$
$$= \frac{1}{(1-\gamma)^2} \left(\sum_s |d_\xi^\pi(s) - d_{\xi,K}^\pi(s)| \cdot \left\|\sum_a Q^{\pi_\theta}(s, a; z^*) \nabla_\theta \pi_\theta(a|s)\right\|\right)^2$$
$$= \frac{1}{(1-\gamma)^2} \left(\sum_s |d_\xi^\pi(s) - d_{\xi,K}^\pi(s)| \cdot \left\|\sum_a Q^{\pi_\theta}(s, a; z^*) \nabla_\theta \pi_\theta(a|s)\right\|\right)^2$$
$$\le \frac{1}{(1-\gamma)^2} \left(\sum_s |d_\xi^\pi(s) - d_{\xi,K}^\pi(s)| \cdot \max_{\|u\|_\infty \le \frac{\|z^*\|_\infty}{1-\gamma}} \|\nabla_\theta \pi(\cdot|s)u\|\right)^2$$
$$\le \frac{\|\nabla_\theta \pi(\cdot|s)\|_{\infty,2}^2 \cdot \|z^*\|_\infty^2}{(1-\gamma)^4} \left(\sum_s |d_\xi^\pi(s) - d_{\xi,K}^\pi(s)|\right)^2$$
$$\le \frac{C^2 \|z^*\|_\infty^2}{(1-\gamma)^6} \gamma^{2K}.$$

Next, we consider the second (variance) term on the right-hand side of (30). By substituting (14) in for $\nabla_\theta \tilde{V}(\theta; z^*)$ to rewrite it in terms of trajectories $\zeta_i$, we have

$$\mathbb{E}\left[\left\|\nabla_\theta \tilde{V}(\theta; z^*) - \mathbb{E}\left[\nabla_\theta \tilde{V}(\theta; z^*)\right]\right\|^2\right] = \frac{1}{n} \mathbb{E}\left[\left\|\nabla_\theta \tilde{V}(\theta; z^*; \zeta_i) - \mathbb{E}\left[\nabla_\theta \tilde{V}(\theta; z^*; \zeta_i)\right]\right\|^2\right]$$
$$\le \frac{1}{n} \mathbb{E}\left[\left\|\nabla_\theta \tilde{V}(\theta; z^*; \zeta_i)\right\|^2\right]$$
$$\le \frac{C^2 \|z^*\|_\infty^2}{n(1-\gamma)^4}.$$

The first inequality comes from crudely upper-bounding the bias by the estimator itself. The last equality uses (27).

Now, returning focus to the second term in the bound (29), by the linearity of the stochastic estimators with respect to the differential and (28), we have

$$\left\|\nabla_\theta \tilde{V}(\theta; z^*) - \nabla_\theta \tilde{V}(\theta; \hat{z})\right\|^2 = \left\|\nabla_\theta \tilde{V}(\theta; z^* - \hat{z})\right\|^2 \le \frac{C^2 \|z^* - \hat{z}\|_\infty^2}{(1-\gamma)^4}.$$

Taking the expectation after squaring both sides yields

$$\mathbb{E}\left[\left\|\nabla_\theta \tilde{V}(\theta; z^*) - \nabla_\theta \tilde{V}(\theta; \hat{z})\right\|^2\right] \leq \frac{C^2}{(1-\gamma)^4} \mathbb{E}\left[\|z^* - \hat{z}\|_\infty^2\right]. \tag{32}$$

Combining inequalities (29), (30), (31), (32), (32) yields

$$\mathbb{E}\left[\|x^* - \hat{x}\|^2\right] \leq \frac{2C^2\|z^*\|_\infty^2}{(1-\gamma)^4} \cdot \left(\frac{\gamma^{2K}}{(1-\gamma)^2} + \frac{1}{n}\right) + \frac{2C^2}{(1-\gamma)^4} \cdot \mathbb{E}\left[\|z^* - \hat{z}\|_\infty^2\right].$$

which is as stated in Lemma 1(ii). $\qquad\square$

### E.4   Proof of Lemma 1(iii).

In this section we will apply the generalization bound for stochastic saddle points from [54] to bound the term $\mathbb{E}[\|\hat{z} - z^*\|_\infty^2]$. To achieve this, we need a compact feasible region for $x$. Note that for problems (25) and (26), the solutions $x^*$ and $\hat{x}$ has the form

$$x^* = \nabla_\theta V(\theta; z^*) \qquad \text{and} \qquad \hat{x} = \nabla_\theta \tilde{V}(\theta; \hat{z}).$$

Due to (28) and the constraint that $\|z\|_\infty \leq \ell_F$, we have $\|x^*\| \leq \frac{C\|z^*\|_\infty}{(1-\gamma)^2} \leq \frac{C\ell_F}{(1-\gamma)^2}$ and thus $\|\hat{x}\| \leq \frac{C\ell_F}{(1-\gamma)^2}$ with probability 1. Therefore, adding a constraint that $\|x\| \leq \frac{C\ell_F}{(1-\gamma)^2}$ will not change the solutions of problems (25) and (26). Formally speaking, we will then apply the theory of [54] to the following pair of constrained problems:

$$(x^*, z^*) = \operatorname*{argmax}_{x \in \mathcal{X}} \operatorname*{argmin}_{z \in \mathcal{Z}} V(\theta; z) + \delta \nabla_\theta V(\theta; z)^\top x - F^*(z) - \frac{\delta}{2}\|x\|^2, \tag{33}$$

and

$$(\hat{x}, \hat{z}) = \operatorname*{argmax}_{x \in \mathcal{X}} \operatorname*{argmin}_{z \in \mathcal{Z}} \tilde{V}(\theta; z) + \delta \nabla_\theta \tilde{V}(\theta; z)^\top x - F^*(z) - \frac{\delta}{2}\|x\|^2. \tag{34}$$

with $\mathcal{X} = \{x : \|x\| \leq \frac{C\ell_F}{(1-\gamma)^2}\}$ and $\mathcal{Z} = \{z : \|z\|_\infty \leq \ell_F\}$. The problems (25) and (33) share the same solution, and problems (26) and (34) share the same solution.

Finally, similar to the proof of (31), for any $x \in \mathcal{X}$ and $z \in \mathcal{Z}$

$$V(\theta; z) + \delta \nabla_\theta V(\theta; z)^\top x - \mathbb{E}\left[\tilde{V}(\theta; z; \zeta_i) + \delta \nabla_\theta \tilde{V}(\theta; z; \zeta_i)^\top x\right] = \mathcal{O}\left(\frac{\gamma^K}{1-\gamma}\right).$$

For the simplicity of discussion, let us assume that $K$ is large enough so that we can ignore the $\mathcal{O}\left(\frac{\gamma^K}{1-\gamma}\right)$ bias. Therefore problem (34) can be viewed as an empirical version of the problem (33) with negligible bias. To apply the theory of [54], define

$$\Psi_\zeta(x, z) := \tilde{V}(\theta; z; \zeta) + \delta \nabla_\theta \tilde{V}(\theta; z; \zeta)^\top x - F^*(z) - \frac{\delta}{2}\|x\|^2.$$

Then for any sample path $\zeta$, $\Psi_\zeta$ satisfies the following set of properties:

- $\Psi_\zeta(\cdot, z)$ is $\mu_x$-strongly concave under $L_2$ norm. And $\Psi_\zeta(x, \cdot)$ is $\mu_z$-strongly convex under the $L_\infty$ norm. In other words, for $\forall x, x' \in \mathcal{X}$ and $z, z' \in \mathcal{Z}$,

$$\begin{cases} \Psi_\zeta(x', z) \geq \Psi_\zeta(x, z) + \langle u, x' - x\rangle + \frac{\mu_x}{2}\|x' - x\|^2, & u \in \partial_x \Psi_\zeta(x, z), \\ \Psi_\zeta(x, z') \leq \Psi_\zeta(x, z) + \langle v, z' - z\rangle - \frac{\mu_z}{2}\|z' - z\|_\infty^2, & v \in \partial_z \Psi_\zeta(x, z). \end{cases}$$

  In our case, it is clear that $\mu_x = \delta$. Due to Theorem 3 of [21], $\mu_z = L_F^{-1}$.

- The feasible regions $\mathcal{X}$ and $\mathcal{Z}$ are compact convex sets. For every $\zeta$, there exist constants $\ell_x(\xi, z)$ and $\ell_z(\xi, x)$ s.t.

$$\begin{cases} |\Psi_\zeta(x', z) - \Psi_\zeta(x, z)| \leq \ell_x(\zeta, z)\|x' - x\|, & \forall x, x' \in \mathcal{X} \text{ and } y \in \mathcal{Y}, \\ |\Psi_\zeta(x, z') - \Psi_\zeta(x, z)| \leq \ell_z(\zeta, x)\|z' - z\|_\infty, & \forall z, z' \in \mathcal{Z} \text{ and } x \in \mathcal{X}. \end{cases}$$

  In our case, we gave $\ell_z(\zeta, x) = \sup\{\|u\|_1 : z \in \mathcal{Z}, u \in \partial_z \Psi_\zeta(x, z)\} = \frac{1}{1-\gamma} + \ell_{F^*} + \mathcal{O}(\delta)$ and $\ell_x(\zeta, z) = \sup_{x \in \mathcal{X}} \|\nabla_x \Psi_\zeta(x, z)\| = \mathcal{O}(\delta)$. Consequently,

$$\begin{cases} (\ell_x^w)^2 := \sup_{z \in \mathcal{Z}} \mathbb{E}\left[\ell_x^2(\zeta, z)\right] = \mathcal{O}(\delta^2), \\ (\ell_z^w)^2 := \sup_{x \in \mathcal{X}} \mathbb{E}\left[\ell_z^2(\zeta, x)\right] = \mathcal{O}(\ell_{F^*}^2 + \frac{1}{(1-\gamma)^2} + \delta^2). \end{cases}$$

With the above two properties, Theorem 1 of [54] indicates that

$$\frac{\mu_z}{2} \mathbb{E}\left[\|\hat{z} - z^*\|_\infty^2\right] \leq \frac{2\sqrt{2}}{n} \cdot \left(\frac{(\ell_x^w)^2}{\mu_x} + \frac{(\ell_z^w)^2}{\mu_z}\right).$$

With the detailed parameters substituted in the above inequality, we have

$$\mathbb{E}\left[\|\hat{z} - z^*\|_\infty^2\right] \leq \mathcal{O}\left(\frac{L_F^2}{n(1-\gamma)^2} + \frac{L_F^2 \ell_{F^*}^2}{n} + \frac{L_F^2 \delta^2 + L_F \delta}{n}\right)$$

as stated in Lemma 1(iii). $\qquad\square$

## F    Proof of Theorem 3

*Proof.* Let $\theta^*$ be a first-order stationary solution of (10). When $F$ is concave and locally Lipschitz continuous in a neighborhood containing $\lambda(\Theta)$, we can compute the Fréchet superdifferential of $F \circ \lambda$ at $\theta^*$ by the chain rule, see [15]. That is

$$\hat{\partial}(F \circ \lambda)(\theta^*) = [\nabla_\theta \lambda(\theta^*)]^\top \partial F(\lambda^*)$$

where $\partial F(\lambda^*)$ denotes the set of supergradients of the concave function $F$ at $\lambda^*$. Then there exists $w^* \in \partial F(\lambda^*) \in \mathbb{R}^{SA}$ such that $u^* := [\nabla_\theta \lambda(\theta^*)]^\top w^* \in \hat{\partial}(F \circ \lambda)(\theta^*)$ as in (17). It follows from (17) that

$$\langle w^*, \nabla_\theta \lambda(\theta^*)(\theta - \theta^*)\rangle \leq 0, \quad \text{for} \quad \forall \theta \in \Theta. \tag{35}$$

For any $\lambda \in \lambda(\Theta)$, we let $\theta := g(\lambda)$ such that $\lambda = \lambda(\theta)$. Therefore, by adding and subtracting $\nabla_\theta \lambda(\theta^*)\theta$ inside the inner product we have

$$
\begin{aligned}
\langle w^*, \lambda - \lambda^*\rangle &= \langle w^*, \lambda(\theta) - \lambda(\theta^*)\rangle \tag{36}\\
&= \langle w^*, \nabla_\theta \lambda(\theta^*)(\theta - \theta^*)\rangle + \langle w^*, \lambda(\theta) - \lambda(\theta^*) - \nabla_\theta \lambda(\theta^*)(\theta - \theta^*)\rangle \\
&\leq 0 + \|w^*\|\|\lambda(\theta) - \lambda(\theta^*) - \nabla_\theta \lambda(\theta^*)(\theta - \theta^*)\|.
\end{aligned}
$$

where in the last inequality we group terms and apply Cauchy-Schwartz. Note that the Jacobian matrix $\nabla_\theta \lambda(\theta)$ is Lipschitz continuous. Denote the Lipschitz constant by $L_\lambda$, i.e., $\|\nabla_\theta \lambda(\theta) - \nabla_\theta \lambda(\theta')\| \leq L_\lambda \|\theta - \theta'\|$ for all $\theta, \theta' \in \Theta$. Then,

$$\|\lambda(\theta) - \lambda(\theta^*) - \nabla_\theta \lambda(\theta^*)(\theta - \theta^*)\| \leq \frac{L_\lambda}{2}\|\theta - \theta^*\|^2.$$

By Assumption 1, we know

$$\|\theta - \theta^*\|^2 = \|g(\lambda) - g(\lambda^*)\|^2 \leq \ell_\theta^2\|\lambda - \lambda^*\|^2.$$

Substituting the above inequalities into (36) yields

$$\langle w^*, \lambda - \lambda^*\rangle \leq \frac{L_\lambda \ell_\theta^2}{2}\|w^*\|\|\lambda - \lambda^*\|^2 \qquad \forall \lambda \in \lambda(\Theta). \tag{37}$$

Note that (37) holds for arbitrary $\lambda \in \lambda(\Theta)$. Therefore, since $\lambda(\Theta)$ is assumed to be convex (Assumption 1(i)), we can also substitute $\lambda$ with $(1-\alpha)\lambda^* + \alpha\lambda, \alpha \in [0,1]$ into the above equation, which yields

$$\alpha\langle w^*, \lambda - \lambda^*\rangle \leq \frac{L_\lambda \ell_\theta^2 \alpha^2}{2}\|w^*\|\|\lambda - \lambda^*\|^2 \qquad \forall \lambda \in \mathcal{L}, \forall \alpha \in [0,1].$$

Divide both sides of the preceding expression by $\alpha$ and take $\alpha \to 0+$ gives

$$\langle w^*, \lambda - \lambda^*\rangle \leq \lim_{\alpha \to 0+} \frac{L_\lambda \ell_\theta^2 \alpha}{2}\|w^*\|\|\lambda - \lambda^*\|^2 = 0 \qquad \forall \lambda \in \lambda(\Theta).$$

Recall that the following problem is concave in $\lambda$:

$$\max_\lambda \quad F(\lambda) \qquad \text{s.t.} \qquad \lambda \in \lambda(\Theta),$$

therefore we conclude that $\lambda^*$ is the global optimal solution. Then $\theta^* = g(\lambda^*)$ is the globally optimal solution of the nonconvex optimization problem (10). $\qquad\square$

# G  Proof of Theorem 4

## G.1  Proof of sublinear convergence

*Proof.* First, the Lipschitz continuity in Assumption 2 indicates that

$$\left| F(\lambda(\theta)) - F(\lambda(\theta^k)) - \langle \nabla_\theta F(\lambda(\theta^k)), \theta - \theta^k \rangle \right| \le \frac{L}{2} \|\theta - \theta^k\|^2.$$

Consequently, for any $\theta \in \Theta$ we have the ascent property:

$$F(\lambda(\theta)) \ge F(\lambda(\theta^k)) + \langle \nabla_\theta F(\lambda(\theta^k)), \theta - \theta^k \rangle - \frac{L}{2} \|\theta - \theta^k\|^2 \ge F(\lambda(\theta)) - L\|\theta - \theta^k\|^2. \quad (38)$$

The optimality condition in the policy update rule (16) then yields

$$\begin{aligned}
F(\lambda(\theta^{k+1})) &\ge F(\lambda(\theta^k)) + \langle \nabla_\theta F(\lambda(\theta^k)), \theta^{k+1} - \theta^k \rangle - \frac{L}{2} \|\theta^{k+1} - \theta^k\|^2 \\
&= \max_{\theta \in \Theta} F(\lambda(\theta^k)) + \langle \nabla_\theta F(\lambda(\theta^k)), \theta - \theta^k \rangle - \frac{L}{2} \|\theta - \theta^k\|^2 \\
&\overset{(a)}{\ge} \max_{\theta \in \Theta} F(\lambda(\theta)) - L\|\theta - \theta^k\|^2 \\
&\overset{(b)}{\ge} \max_{\alpha \in [0,1]} \left\{ F(\lambda(\theta_\alpha)) - L\|\theta_\alpha - \theta^k\|^2 : \theta_\alpha = g(\alpha\lambda(\theta^*) + (1-\alpha)\lambda(\theta^k)) \right\}. \quad (39)
\end{aligned}$$

Here, step (a) is due to (38) and step (b) uses the convexity of $\lambda(\Theta)$. Now, we proceed to analyze the right-hand side of (39). First, by the concavity of $F$ and the fact that $\lambda \circ g = id$, we have

$$F(\lambda(\theta_\alpha)) = F(\alpha\lambda(\theta^*) + (1-\alpha)\lambda(\theta^k)) \ge \alpha F(\lambda(\theta^*)) + (1-\alpha)F(\lambda(\theta^k)).$$

Moreover, by the Lipschitz continuity assumption of $g$, we have

$$\begin{aligned}
\|\theta_\alpha - \theta^k\|^2 &= \|g(\alpha\lambda(\theta^*) + (1-\alpha)\lambda(\theta^k)) - g(\lambda(\theta^k))\|^2 \quad (40) \\
&\le \alpha^2 \ell_\theta^2 \|\!|\lambda(\theta^*) - \lambda(\theta^k)|\!\|^2 \\
&\le \alpha^2 \ell_\theta^2 D_\lambda^2.
\end{aligned}$$

Substituting the above two inequalities into the right-hand side of (39), we get

$$\begin{aligned}
F(\lambda(\theta^*)) &- F(\lambda(\theta^{k+1})) \\
&\le \min_{\alpha \in [0,1]} \left\{ F(\lambda(\theta^*)) - F(\lambda(\theta_\alpha)) + L\|\theta_\alpha - \theta^k\|^2 : \theta_\alpha = g(\alpha\lambda(\theta^*) + (1-\alpha)\lambda(\theta^k)) \right\} \\
&\le \min_{\alpha \in [0,1]} (1-\alpha)\left( F(\lambda(\theta^*)) - F(\lambda(\theta^k)) \right) + \alpha^2 L\ell_\theta^2 D_\lambda^2. \quad (41)
\end{aligned}$$

Let $\alpha_k = \frac{F(\Lambda(\pi^*)) - F(\Lambda(\pi^k))}{2L\ell_\theta^2 D_\lambda^2} \ge 0$, which is the minimizer of the RHS of (41) as long as it satisfies $\alpha_k \le 1$.

Now, we claim the following: If $\alpha_k \ge 1$ then $\alpha_{k+1} < 1$. Further, if $\alpha_k < 1$ then $\alpha_{k+1} \le \alpha_k$. The two claims together mean that $(\alpha_k)_k$ is decreasing and all $\alpha_k$ are in $[0,1)$ except perhaps $\alpha_0$.

To prove the first of the two claims, assume $\alpha_k \ge 1$. This implies that $F(\Lambda(\pi^*)) - F(\Lambda(\pi^k)) \ge 2L\ell_\theta^2 D_\lambda^2$. Hence, choosing $\alpha = 1$ in (41), we get

$$F(\lambda(\theta^*)) - F(\lambda(\theta^k)) \le L\ell_\theta^2 D_\lambda^2$$

which implies that $\alpha_{k+1} \le 1/2 < 1$.

To prove the second claim, we plug $\alpha_k$ into (41) to get

$$F(\lambda(\theta^*)) - F(\lambda(\theta^{k+1})) \le \left( 1 - \frac{F(\lambda(\theta^*)) - F(\lambda(\theta^k))}{4L\ell_\theta^2 D_\lambda^2} \right) (F(\lambda(\theta^*)) - F(\lambda(\theta^k))),$$

which shows that $\alpha_{k+1} \le \alpha_k$ as required.

Now, by our preceding discussion, for $k = 1, 2, \ldots$ the previous recursion holds. Using the definition of $\alpha_k$, we rewrite this in the equivalent form

$$\frac{\alpha_{k+1}}{2} \leq \left(1 - \frac{\alpha_k}{2}\right) \cdot \frac{\alpha_k}{2}.$$

By rearranging the preceding expressions and algebraic manipulations, we obtain

$$\frac{2}{\alpha_{k+1}} \geq \frac{1}{\left(1 - \frac{\alpha_k}{2}\right) \cdot \frac{\alpha_k}{2}} = \frac{2}{\alpha_k} + \frac{1}{1 - \frac{\alpha_k}{2}} \geq \frac{2}{\alpha_k} + 1.$$

For simplicity assume that $\alpha_0 < 1$ also holds. Then, $\frac{2}{\alpha_k} \geq \frac{2}{\alpha_0} + k$, and consequently

$$F(\lambda(\theta^*)) - F(\lambda(\theta^k)) \leq \frac{F(\lambda(\theta^*)) - F(\lambda(\theta^0))}{1 + \frac{F(\lambda(\theta^*)) - F(\lambda(\theta^0))}{4L\ell_\theta^2 D_\lambda^2} \cdot k} \leq \frac{4L\ell_\theta^2 D_\lambda^2}{k}.$$

A similar analysis holds when $\alpha_0 > 1$. Combining these two gives that $F(\lambda(\pi^*)) - F(\lambda(\pi^k)) \leq \frac{4L\ell_\theta^2 D_\lambda^2}{k+1}$ no matter the value of $\alpha_0$, which proves the result. $\qquad \square$

### G.2   Proof of exponential convergence

When the strong concavity of $F$ is available, we further provide the exponential convergence result.

*Proof.* We start from (39) whose proof requires no assumption on strong concavity of $F$, which is

$$F(\lambda(\theta^{k+1})) \geq \max_{\alpha \in [0,1]} \left\{ F(\lambda(\theta_\alpha)) - L\|\theta_\alpha - \theta^k\|^2 : \theta_\alpha = g(\alpha\lambda(\theta^*) + (1-\alpha)\lambda(\theta^k)) \right\}. \quad (42)$$

By the $\mu$-strong concavity of $F$, we have

$$F(\lambda(\theta_\alpha)) = F(\alpha\lambda(\theta^*) + (1-\alpha)\lambda(\theta^k)) \geq \alpha F(\lambda(\theta^*)) + (1-\alpha)F(\lambda(\theta^k)) + \frac{\mu}{2}\alpha(1-\alpha)\|\|\lambda(\theta^*) - \lambda(\theta^k)\|\|^2.$$

By the Lipschitz continuity of $g$, we know that

$$\|\theta_\alpha - \theta^k\| = \|g(\alpha\lambda(\theta^*) + (1-\alpha)\lambda(\theta^k)) - g(\lambda(\theta^k))\| \leq \alpha\ell_\theta\|\|\lambda(\theta^*) - \lambda(\theta^k)\|\|$$

Substituting the above two inequalities into the right-hand side of (42), we get

$$F(\lambda(\theta^*)) - F(\lambda(\theta^{k+1})) \quad (43)$$
$$\leq \min_{\alpha \in [0,1]} \left\{ F(\lambda(\theta^*)) - F(\lambda(\theta_\alpha)) + L\|\theta_\alpha - \theta^k\|^2 : \theta_\alpha = g(\alpha\lambda(\theta^*) + (1-\alpha)\lambda(\theta^k)) \right\}$$
$$\leq \min_{\alpha \in [0,1]} (1-\alpha)\left(F(\lambda(\theta^*)) - F(\lambda(\theta^k))\right) - \alpha\left(\frac{1-\alpha}{2}\mu - L\ell_\theta^2\alpha\right)\|\|\lambda(\theta^*) - \lambda(\theta^k)\|\|^2$$

Suppose we choose $\bar{\alpha} = \frac{1}{1 + L\ell_\theta^2/\mu} < 1$ such that $\left(\frac{1-\bar{\alpha}}{2}\mu - L\ell_\theta^2\bar{\alpha}\right) = 0$. Then we have a contraction with modulus $1 - \bar{\alpha}$ as

$$F(\lambda(\theta^*)) - F(\lambda(\theta^{k+1})) \leq (1-\bar{\alpha})F(\lambda(\theta^*)) - F(\lambda(\theta^k)).$$

Consequently, for any $k \geq 1$, we have

$$F(\lambda(\theta^*)) - F(\lambda(\theta^k)) \leq (1-\bar{\alpha})^k \left(F(\lambda(\theta^*)) - F(\lambda(\theta^0))\right).$$

which can be translated into iteration complexity by fixing $\epsilon$ and initialization $\theta^0$, and solving for the minimal $k$ such that $F(\lambda(\theta^*)) - F(\lambda(\theta^k)) \leq \epsilon$. Doing so is an algebraic exercise which results in

$$\mathcal{O}\left(\frac{1}{\bar{\alpha}}\log\left(\frac{F(\lambda(\theta^*)) - F(\lambda(\theta^0))}{\epsilon}\right)\right) = \mathcal{O}\left(\frac{L\ell_\theta^2}{\mu}\log\left(\frac{1}{\epsilon}\right)\right)$$

$\qquad \square$

## H Validating Assumption 1 for tabular policy case

For the tabular policy case, the following Proposition holds true and hence the Assumption 1 is satisfied in this case.

**Proposition 1.** *Suppose $\xi_s > 0$ for $\forall s \in \mathcal{S}$. Then the following hold:*

*(i). The mappings $\Pi$ and $\Lambda$ form a pair of bijections between the convex sets $\Delta_{\mathcal{A}}^{\mathcal{S}}$ and $\mathcal{L}$;*

*(ii). $\exists L_\lambda > 0$ s.t. $\|\nabla\Lambda(\pi) - \nabla\Lambda(\pi')\| \le L_\lambda \|\pi - \pi'\|, \forall \pi, \pi' \in \Delta_{\mathcal{A}}^{\mathcal{S}}$;*

*(iii). For all $\lambda, \lambda' \in \mathcal{L}$, we have*

$$\|\Pi(\lambda) - \Pi(\lambda')\|^2 \le 2 \sum_s \Big( \sum_a (\lambda'_{sa} - \lambda_{sa})^2 + (\sum_a \lambda'_{sa} - \lambda_{sa})^2 \Big) / \Big( \sum_a \lambda_{sa} \Big)^2.$$

*Consequently, $\|\Pi(\lambda) - \Pi(\lambda')\| \le \frac{2}{\min_s \xi_s} \|\lambda - \lambda'\|_1$*

*Proof.*
**Proof of (i):** The equations $\Pi \circ \Lambda = \mathrm{id}_{\mathcal{L}}$ and $\Lambda \circ \Pi = \mathrm{id}_{\Delta_{\mathcal{A}}^{\mathcal{S}}}$ are standard. See, e.g., [3] or Appendix A of [53].
**Proof of (ii):** For the existence of the $L_\lambda$-Lipschitz constant of the gradient $\nabla\Lambda$, note that the $t$-th term of the infinite sum

$$\Lambda_{sa}(\pi) = \sum_{t=0}^{\infty} \gamma^t \cdot \mathbb{P}\Big( s_t = s, a_t = a \ \Big| \ \pi, s_0 \sim \xi \Big)$$

is a $(t+1)$-th order polynomial. Therefore, $\Lambda_{sa}(\pi)$ can actually be defined for any $\pi$ even if $\pi \notin \Delta_{\mathcal{A}}^{\mathcal{S}}$, as long as this infinite series of polynomial of $\pi$ converges absolutely. Note that for $\forall \pi \in \Delta_{\mathcal{A}}^{\mathcal{S}}$, since $0 \le \mathbb{P}\big( s_t = s, a_t = a \ \big| \ \pi, s_0 \sim \xi \big) \le 1$ this infinite series is absolutely convergent. Because we have $0 < \gamma < 1$, even if we slightly perturb the $\pi$ within a neighborhood of it (not necessarily in $\Delta_{\mathcal{A}}^{\mathcal{S}}$ after perturbation), the infinite series is still absolutely convergent. This indicates that $\Lambda_{sa}$ is infinitely continuously differentiable in an open neighborhood containing $\Delta_{\mathcal{A}}^{\mathcal{S}}$, then due to the compactness of $\Delta_{\mathcal{A}}^{\mathcal{S}}$, we are able to argue that there exists a $L_\lambda$ s.t. $\nabla\Lambda$ is $L_\lambda$-Lipschitz continuous within $\Delta_{\mathcal{A}}^{\mathcal{S}}$.

**Proof of (iii):** Now, we provide the calculation of the Lipschitz constant of $\Pi$. For the ease of notation, let us define $\mu_s = \sum_{a \in \mathcal{A}} \lambda_{sa}$ and $\mu'_s = \sum_{a \in \mathcal{A}} \lambda'_{sa}$. Then for $\forall \lambda, \lambda' \in \mathcal{L}$ and $\forall (s, a) \in \mathcal{S} \times \mathcal{A}$, it holds that

$$
\begin{aligned}
\Pi_{sa}(\lambda) - \Pi_{sa}(\lambda') &= \frac{\lambda_{sa}}{\mu_s} - \frac{\lambda'_{sa}}{\mu'_s} \\
&= \Big( \frac{\lambda_{sa}}{\mu_s} - \frac{\lambda'_{sa}}{\mu_s} \Big) + \Big( \frac{\lambda'_{sa}}{\mu_s} - \frac{\lambda'_{sa}}{\mu'_s} \Big) \\
&= \frac{1}{\mu_s}(\lambda_{sa} - \lambda'_{sa}) + \frac{\mu'_s - \mu_s}{\mu_s \mu'_s} \lambda'_{sa}.
\end{aligned}
$$

Consequently, we can compute the norm difference of the preceding expression and apply the triangle inequality:

$$
\begin{aligned}
\|\Pi(\lambda) - \Pi(\lambda')\|^2 &= \sum_{s \in \mathcal{S}} \sum_{a \in \mathcal{A}} \big( \Pi_{sa}(\lambda) - \Pi_{sa}(\lambda') \big)^2 \qquad (44) \\
&\le 2 \sum_{s \in \mathcal{S}} \sum_{a \in \mathcal{A}} \frac{1}{\mu_s^2} (\lambda_{sa} - \lambda'_{sa})^2 + 2 \sum_{s \in \mathcal{S}} \sum_{a \in \mathcal{A}} \frac{(\mu'_s - \mu_s)^2}{\mu_s^2 (\mu'_s)^2} (\lambda'_{sa})^2 \\
&\le 2 \sum_{s \in \mathcal{S}} \frac{1}{\mu_s^2} \Big( \sum_{a \in \mathcal{A}} (\lambda_{sa} - \lambda'_{sa})^2 + (\mu'_s - \mu_s)^2 \Big),
\end{aligned}
$$

where the last inequality follows because $\|x\|_2^2 \leq \|x\|_1^2$ holds for any vector $x$ (here, $\|\cdot\|_p$ denotes the $p$-norm). Finally, note that $\mu_s \geq \xi_s > 0$, we have

$$
\begin{aligned}
\|\Pi(\lambda) - \Pi(\lambda')\|^2 &\leq 2\sum_{s\in\mathcal{S}}\frac{1}{\mu_s^2}\left(\sum_{a\in\mathcal{A}}(\lambda_{sa}-\lambda'_{sa})^2 + (\mu'_s - \mu_s)^2\right) \\
&\leq \frac{2}{\min_s \xi_s^2}\sum_{s\in\mathcal{S}}\left(\sum_{a\in\mathcal{A}}(\lambda_{sa}-\lambda'_{sa})^2 + \left(\sum_{a\in\mathcal{A}}|\lambda_{sa}-\lambda_{sa'}|\right)^2\right) \\
&\leq \frac{4}{\min_s \xi_s^2}\|\lambda - \lambda'\|_1^2
\end{aligned}
$$

Take the square root of both sides completes the proof. $\qquad\square$

## I   Proof of Theorem 5

*Proof.* To prove this theorem, it suffices to observe that (39) is still true with $\theta = \pi$, $\lambda(\theta) = \Lambda(\pi)$ and $g(\lambda) = \Pi(\lambda)$. Therefore, (39) can be translated as

$$
F(\Lambda(\pi^{k+1})) \geq \max_{\alpha\in[0,1]}\left\{F(\Lambda(\pi_\alpha)) - L\|\pi_\alpha - \pi^k\|^2 : \pi_\alpha = \Pi(\alpha\Lambda(\pi^*) + (1-\alpha)\Lambda(\pi^k))\right\}. \quad (45)
$$

By the concavity of $F$ and the fact that $\Lambda \circ \Pi = id$, we have

$$
F(\Lambda(\pi_\alpha)) = F(\alpha\Lambda(\pi^*) + (1-\alpha)\Lambda(\pi^k)) \geq \alpha F(\Lambda(\pi^*)) + (1-\alpha)F(\Lambda(\pi^k)). \quad (46)
$$

For the inequality (40), we can derive a tighter bound by the following argument:

$$
\begin{aligned}
\|\pi_\alpha - \pi^k\|^2 &= \|\Pi(\alpha\Lambda(\pi^*) + (1-\alpha)\Lambda(\pi^k)) - \Pi(\Lambda(\pi^k))\|^2 \quad (47)\\
&\leq \alpha^2 \sum_s \frac{1}{\left(\sum_a \lambda_{sa}\right)^2}\left(\sum_a(\lambda^*_{sa}-\lambda_{sa})^2 + \left(\sum_a\lambda^*_{sa} - \sum_a\lambda_{sa}\right)^2\right) \\
&\leq 4\alpha^2 \sum_s \frac{1}{\left(\sum_a \lambda_{sa}\right)^2}\left(\left(\sum_a\lambda^*_{sa}\right)^2 + \left(\sum_a\lambda_{sa}\right)^2\right) \\
&= 4\alpha^2 \sum_s \frac{\left(d^{\pi^*}_\xi(s)\right)^2 + \left(d^{\pi^k}_\xi(s)\right)^2}{\left(d^{\pi^k}_\xi(s)\right)^2} \\
&= 4\alpha^2|\mathcal{S}| + 4\alpha^2 \sum_s \left(\frac{d^{\pi^*}_\xi(s)}{d^{\pi^k}_\xi(s)}\right)^2 \\
&\leq 4\alpha^2|\mathcal{S}| + 4\alpha^2|\mathcal{S}|\left\|\frac{d^{\pi^*}_\xi}{d^{\pi^k}_\xi}\right\|_\infty^2 \\
&\leq 4\alpha^2|\mathcal{S}|\cdot\left(1 + (1-\gamma)^{-2}\left\|d^{\pi^*}_\xi/\xi\right\|_\infty^2\right) \\
&\leq \frac{5\alpha^2|\mathcal{S}|}{(1-\gamma)^2}\left\|d^{\pi^*}_\xi/\xi\right\|_\infty^2
\end{aligned}
$$

Denote $D := \frac{5|\mathcal{S}|}{(1-\gamma)^2}\left\|d^{\pi^*}_\xi/\xi\right\|_\infty^2$. Substituting the above two inequalities into the right-hand side of (45), we get

$$
\begin{aligned}
F(\Lambda(\pi^*)) &- F(\Lambda(\pi^{k+1})) \\
&\leq \min_{\alpha\in[0,1]}\left\{F(\Lambda(\pi^*)) - F(\Lambda(\pi_\alpha)) + L\|\pi_\alpha - \pi^k\|^2 : \pi_\alpha = \Pi(\alpha\Lambda(\pi^*) + (1-\alpha)\Lambda(\pi^k))\right\} \\
&\leq \min_{\alpha\in[0,1]}(1-\alpha)\left(F(\Lambda(\pi^*)) - F(\Lambda(\pi^k))\right) + LD\alpha^2. \quad (48)
\end{aligned}
$$

Note that (48) differs from (41) by replacing $\ell_\theta^2 D_\lambda^2$ with $D$. The latter proof of Theorem 5 is almost identical to that of Theorem 4 and hence we omit the proof. $\qquad\square$