[Reviews · NeurIPS 2020]

Review 1

Summary and Contributions: The paper studies policy gradient methods for optimizing the concave functions of the state occupancy measure. It gives a simple, clear, argument for why such a method reaches a global optimum when the parameterized policy class contains all policies. The paper also provides a formula for the gradient in terms of value function space. A couple of numerical experiments are conducted in a simple environment.

Strengths: Theorem 3 is simple (which is good!) and clear. There is a lot of conceptual value to the community in this perspective and in the observation that it allows one to optimize more general objectives. The gradient estimation procedure in Theorem 1 also took creativity.

Weaknesses: Theorem 13, while creative, has some major weaknesses. It can hold in tabular cases, where every possible policy is in the policy class. But it's not clear if there are any other meaningful examples. (Certainly, it requires that the policy class is convex, but since the mapping \pi \mapsto \lambda(\pi) is nonconvex, it requires much more than that) . Assumption 1 also rules out softmax policies. It effectively requires the Jacobian matrix has eigenvalues bounded above and below by constants. It also requires \Theta is closed. These fail for softmax policies (the derivatives can vanish, the paraemter space is unbounded). The authors should emphasize this. I don't see how Theorem 1 would be applied in settings with an extremely large state space.

Correctness: The claims look correct, but other than proof of Thm 1, I did not read the proofs in the appendix.

Clarity: The writing is fine.

Relation to Prior Work: The paper should highlight that a "dual" or saddle-point perspective on the classical policy gradient theorem is not new. See for example this introductory paper: "Reinforcement Learning via Fenchel-Rockafellar Duality." [18] seems to treat general problems with concave objectives and not just maximum entropy exploration. The authors should describe why they feel their approach offers advantages.

Reproducibility: Yes

Additional Feedback:


Review 2

Summary and Contributions: The paper addresses the problem of extending the cumulative rewards objective with more general utilities, such as any concave function of the state-action occupancies. Especially, it presents a Variational Policy Gradient Theorem for the general setting, which casts the policy gradient as the solution of a stochastic saddle-point problem. It further provides (1) a principled method to estimate the newly introduced policy gradient, (2) a convergence analysis of the corresponding policy optimization process, and (3) a brief numerical validation.

Strengths: I found extremely relevant the idea of providing a theoretical understanding of RL beyond reward maximization, which also provides a compelling unifying perspective for a large body of work in the still growing threads of risk-averse RL, pure exploration, and imitation learning. The formulation of the Variational Policy Gradient is really nice, even if it might result too complicated for practical policy optimization. The theoretical results are impressive, considering an error rate that matches the one of traditional policy gradient and a convergence rate of O(1/t), which also improves the best known rate in the special cumulative rewards case (tabular policies), and it upgrades to exponential convergence in the relevant KL-regularized setting.

Weaknesses: My evaluation for the work is mostly positive. I detail below some additional comments and questions that authors might address in their response.

Correctness: I did not check the proofs in detail but the methodology seems sound. The empirical methodology seems also ok, but the results could be reported more clearly (see point 5 below)

Clarity: Yes

Relation to Prior Work: Partly, see points 6 and 7 below

Reproducibility: Yes

Additional Feedback: Update: thanks for the answer, it helped clarify some points. I think the proposed additions will improve the clarity of the paper. ---- 1. While providing a common theoretical ground for general utilities in RL is not a minor contribution by any means, I would have loved to find a discussion on how to build upon these results. Do authors think their work can be leveraged to develop more efficient algorithm in the context of RL with general utilities, or the intended outcome is a deeper understanding of the setting without particular practical upsides? 2. Where the Variational Policy Gradient approach stands in comparison with other policy optimization methods for (specific) general utilities, e.g. CVaR policy optimization [e.g, Chow and Ghavamzadeh, Algorithms for CVaR optimization in MDPs, 2014] or MaxEnt policy optimization [Hazan et al., 2019]? The additional generality imposes any cost in the efficiency of the optimization? 3. Can you comment on the interpretation of the reward term z, which is a quite interesting by-product of the Variational Policy Gradient estimation? What happens if we fully optimize a policy w.r.t. this reward function? How that would compare with the MaxEnt algorithm, which casts policy optimization as a sequence of cumulative rewards problems? 4. The reported results are restricted to stationary Markovian policies. This is a common choice for cumulative rewards objective, since it is well-known that this policy space suffices. Is it also the case for general utilities? 5. I found Figure 1 quite confusing. It is not clear to me if the curves are function of the number of samples, episodes, or iterations. While the estimate for any considered utility converges to the true gradient, estimating the gradient for the entropy objective seems quite inefficient in practice. Oddly enough, outside of the gradient of cumulative rewards, all the estimates point to opposite direction w.r.t. the true gradient at first. 6. In a sense, this work provides a "concretization" of the very abstract framework of [1]. I think the concept of "hidden convexity" is already there, and it is nice to see it finally exploited. You should at least mention [1]. 7. I would like to see a discussion on the "O-REPS" [2] family of algorithms. I am not too familiar with those, but from my understanding they optimize in the space of occupancy measures directly. The main critique (see e.g. [3]) is that they are not practical, since one cannot obtain the optimal policy parameters for general parametric policies. You method seems an improvement in that sense. 8. Typo at line 456: we have we omit -> we omit 9. How gamma and delta terms can be avoided when computing the gradients of z and x in equation (18,19)? [1] Casey Chu, Jose H. Blanchet, Peter W. Glynn: Probability Functional Descent: A Unifying Perspective on GANs, Variational Inference, and Reinforcement Learning. ICML 2019: 1213-1222 [2] Zimin, A. and Neu, G. Online learning in episodic marko-vian decision processes by relative entropy policy search.InAdvances in neural information processing systems,pp. 1583–1591, 2013 [3] Yonathan Efroni, Lior Shani, Aviv Rosenberg, Shie Mannor: Optimistic Policy Optimization with Bandit Feedback. CoRR abs/2002.08243 (2020)


Review 3

Summary and Contributions: The paper solves a fundamental problem where the objectives of RL are general utilities. The authors derive a new variational policy gradient and a variational Monte Carlo gradient estimation method based on sampled paths. Theoretical analysis of the properties and empirical experimental results are also given.

Strengths: This paper develops a new optimization algorithm for reinforcement learning with general utilities. The convergence rate of the new method is also given when the problem satisfies certain convexity. Empirical comparision with policy gradient is given to demenstrate its benefits.

Weaknesses: The proposed variational policy gradient algorithm converges more slowly than RN-Reinforce, which may influence its applications in real setting.

Correctness: The logic is sound and partially supported by experiments.

Clarity: The paper is easy to follow and clearly written.

Relation to Prior Work: Related works are well addressed.

Reproducibility: Yes

Additional Feedback:

[Author Response · NeurIPS 2020]

**Response to Reviewer #2**:

(C1) *Theorem 3 holds in tabular cases, and requires the policy class be convex. It's not clear if there are any other meaningful examples.* Theorem 3 doesn't require the policy class $\pi_\theta$ be convex, but it requires $\Theta$ and $\lambda(\Theta)$ being convex and a bijection between $\Theta$ and $\lambda(\Theta)$. With proper regularity condition on the loss function, one could still manage to prove the result for soft-max policy, which would require a case-specific proof and limiting argument (since it does not satisfy AS1). However, in this paper, we do not wish to complicate the simple and clear form of Theorem 3 as well as its proof.

(C2) *Assumption 1 requires the Jacobian has bounded eigenvalues, thus it fails for softmax policies (the derivatives can vanish). The authors should emphasize this.* Yes, we agree with reviewer's comment. More precisely, we will add the following remark in the revised paper:"It is worth noting that the AS1 implicity requires the minimum singular value of the Jacobian matrix $\nabla\lambda(\cdot)$ to be bounded away from 0 and the convex parameter set $\Theta$ to be compact. The result does hold for tabular soft-max policy if $\Theta$ is restricted to a compact subset of the orthogonal complement of the all-one vectors, but it doesn't hold for general soft-max parameterization unless there is additional regularization. It remains future work to understand the behavior of PG method under a broader family of policy parameterizations."

(C3) *How Theorem 1 would work with an extremely large state space.* Regardless of how large the state space is, the convergence rate of gradient estimates is only determined by properties of $F$ (Theorem 2). To make solving (13) more computationally efficient, one could handle the high-dimensional $z$ using additional/compatible function approximation, though this will induce approximation error (depending on specific choices of $F$) and will require further analysis.

(C4,5) *Cite "Reinforcement Learning via FR Duality." Describe why the paper's approach offers advantages over [18].* Thanks, we will cite the duality paper with discussions. The max-entropy method [18] alternates between density estimation and a planning oracle, and it seems limited to tabular problems and hard to directly work with large state space. In contrast, we focus on understanding the impact of policy parameterization which offers the potential to handle a larger state space, and our work provide a complementary alternative. See also response to Reviewer #3 (C2).

**Response to Reviewer #3**:

(C1) *How to leverage this work to develop more efficient RL algorithms? Or the intended outcome is a deeper understanding of the setting without particular practical upsides?* Both. While we focus on the fundamental optimization theory for RL with general utility, our approach can also yield simpler algorithms for a broad range of RL tasks such as efficient exploration or risk-sensitive policy search. Developing more practically efficient algorithms will require a case-by-case investigation for specific utilities in future work (for example entropy and barrier risk have different properties and probably will need to be handled slightly differently).

(C2) *Compare with CVaR policy optimization [e.g, C&G 2014] or MaxEnt policy optimization [Hazan et al., 2019]?*

C&G 2014 considers cost minimization subject to a CVaR constraint and follows a primal-dual gradient method that uses three timescales. In comparison, our approach exploits the hidden convexity of the CVaR constraint in $\lambda$ and offers an alternative approach. Compared to [Hazan et al., 2019] which focused on tabular MDP and requires a planner oracle, we propose a method of direct policy search that allows parametrization for handling large-scale problems. This makes the setting we consider, as well as our algorithms, more suitable for practical use. See also response to Reviewer #2 (C5).

(C3) *Interpretation of the reward term z.* The entity $z$, instead of being observed directly from the environment, may be interpreted as the "shadow reward" derived via the Fenchel conjugate in Theorem 1. We use the term shadow reward because it plays the algorithmic role of a reward function although it is not. This is similar in spirit to shadow prices in constrained optimization/resource allocation. In a way, our PG estimation algorithm is learning the shadow reward simultaneously while it estimates the gradient.

(C4) *The reported results are restricted to stationary Markovian policies.* This is a common choice for cumulative rewards objective, since it is well-known that this policy space suffices. Is it also the case for general utilities?

Excellent question! Stationary policies are indeed sufficient, because the set of occupancy measures generated by any policy is the same as that of generated by stationary policies [Put14, Hazan etal 19]. Another way to show this is to note that $\max_\pi R(\pi_\theta) = \max_\pi \min_z V(\pi; z) - F^*(z)$. By leveraging the hidden convexity in the $\lambda$ space, strong duality holds between $\lambda, z$, thus one can swap "min" and "max". Then for any fixed $z$ the best-response policy always solves a standard MDP, therefore it suffices to focus on stationary policies and there's zero gap. We will be happy to add this argument to the final paper.

(C5) *Figure 1: the curves are function of the number of samples, episodes, or iterations? Also estimating the gradient for the entropy objective seems quite inefficient in practice.* Thanks for catching. The x-axis is the number of episodes. As for the entropy objective, entropy estimation by drawing samples from a distribution is known to be hard, and even the best estimate converges slowly, therefore it is expected that gradient estimation for this objective also converges slowly.

(C6,7) *Relations to [1,2,3].* Thank you for suggesting these papers. We will add discussions about them. Compared to O-REPS, our algorithm enjoys the fact that they are policy gradient methods and can be implemented flexibly with parametrization, having the potential of being applicable to a larger state space.

(C8) *How gamma and delta terms can be avoided when computing the gradients of z and x in equation (18,19)?*

In (18,19), $\gamma$ is subsumed into terms involving $F^*(z)$ and $Q^{\pi_\theta}$, and $\delta$ is let to go to zero.

**Response to Reviewer #4**: We thank the reviewer for the positive feedback and recommending the paper for acceptance.

[Meta-Review · NeurIPS 2020]

The paper proposes an unifying view on several interesting problems for the RL community (reward maximization, pure-exploration, risk averse RL). It presents a generic Policy Gradient Theorem and studies the convergence of the corresponding policy gradient ascent, which is an important contribution.